# Adaptations in *Plasmodium* tubulin determine distinct microtubule architectures, mechanics and drug susceptibility

Mamata Bangera[1,7], Jiangbo Wu[2], Daniel Beckett[2], Dominik Fachet[3,4], Josie L. Ferreira[5], Gregory A. Voth[2], Simone Reber[3,6] & Carolyn A. Moores[1]

Microtubules are ubiquitous yet diverse cytoskeleton filaments. However, tubulin conservation presents challenges in understanding the origins of diverse microtubule architectures. The mechanisms by which microtubule architecture varies through the life cycle of the malaria-causing parasite *Plasmodium* are not understood and provide a valuable framework for exploring how intrinsic properties of tubulin contribute to architectural variety. Using parasite-purified tubulin, we determine *P. falciparum* microtubule structures by cryo-electron microscopy. Parasite-specific sequences change the tubulin dimer structure, suggesting how drug susceptibility and polymer properties are modified. Within the *P. falciparum* microtubule, lateral contacts are smaller but stronger, and the lattice is stiffer than in brain microtubules. Non-canonical microtubule architectures found in parasites are highly similar to those observed in vitro, validating the physiological relevance of these properties. Our findings show how evolutionary adaptation of tubulin modulates the material properties of the microtubule cytoskeleton.

Microtubules are components of the eukaryotic cytoskeleton involved in numerous cellular activities, including cell division, motility, and specialization. Despite these very different roles, the building blocks of microtubules, tubulin heterodimers composed of α- and β-tubulin, are amongst the most conserved eukaryotic proteins. This presents a major challenge for understanding how microtubules can perform their many functions across the diverse contexts that characterize the eukaryotic tree of life. The specific behaviors of microtubules, including their ability to exhibit GTP-driven dynamic instability, can be modified by microtubule-associated proteins (MAPs), post-translational modifications (PTMs) or tubulin isoforms[1–7]. In addition to influencing dynamics, MAPs, PTMs and isoforms can also affect the cylindrical architecture of microtubules. For example, while many eukaryotic cells contain exclusively 13-protofilament microtubules, a range of other architectures have been reported[8]. However, we don't yet understand how these diverse architectures are specified or what their functional significance in distinct physiological settings is. It is therefore critical to probe the molecular basis of these variations to understand their mechanistic consequences.

[1]Institute of Structural and Molecular Biology, Birkbeck, University of London, London, UK. [2]Department of Chemistry, Chicago Center for Theoretical Chemistry, Institute for Biophysical Dynamics, and James Franck Institute, The University of Chicago, Chicago, IL, USA. [3]Max Planck Institute for Infection Biology, Berlin, Germany. [4]IRI Life Sciences, Humboldt-Universität zu Berlin, Berlin, Germany. [5]Institute of Structural and Molecular Biology, University College, London, UK. [6]Berliner Hochschule für Technik, Berlin, Germany. [7]Present address: Indian Institute of Technology Madras, Chennai, India. ✉e-mail: reber@mpiib-berlin.mpg.de; c.moores@bbk.ac.uk

Recent breakthroughs in cellular imaging have revealed a remarkable diversity of microtubule-based structures in the malaria-causing unicellular parasite, *Plasmodium falciparum*. Throughout its complex life cycle, *P. falciparum* exhibits many different microtubule arrays including dynamic spindle microtubules as well as highly stable subpellicular microtubules (SPMTs) attached to the inner membrane complex of the parasite. Depending on the parasite life cycle stage, variable architectures have been observed for each of its microtubule sub-populations[9]. Such variation is thought to be critical for the parasite to achieve a combination of strength and flexibility as it navigates through fluctuating environments in different hosts. How these different structures emerge from a very small set of tubulin building blocks (two α- and one β-tubulin isoform), often within the same cellular compartment is not well understood. Further, the indispensable role of microtubules in parasite proliferation and infectivity highlights their possibility as potential targets for malaria drug treatment, where the >600,000 fatalities annually, and rising drug resistance to existing treatments are very serious concerns[10]. Understanding the molecular basis of microtubule assembly in *P. falciparum* may provide mechanistic explanations for the actions of chemicals already known to perturb parasite microtubules, as well as enable the structure-guided development of parasite-specific inhibitors.

In this work, we use cryo-electron microscopy (cryo-EM) to determine high-resolution 3D structures of *P. falciparum* microtubules polymerized in different nucleotide states from parasite-isolated tubulin. Our findings demonstrate the innate tendency of *P. falciparum* tubulin to form microtubules with non-canonical protofilament numbers depending on the polymerization conditions. We visualize lattice contacts and ligand-binding sites in these microtubules to understand the structural properties of *P. falciparum* tubulin. A combination of our structural discoveries and molecular dynamics (MD) simulations using experimentally derived models reveals the direct and indirect contribution of parasite-specific sequences to lattice interactions and mechanics. To understand the relationship between parasite and host cytoskeletons, we also compare structural models of *P. falciparum* and well-studied brain tubulin. The observed high degree of conservation of tubulin is not surprising given the central importance of microtubules in eukaryotes. However, our findings show that, because of the polymeric nature of microtubules, distinct parasite cytoskeletal architectures can emerge from small changes in tubulin dimer flexibility and lateral contacts. Our research presents intriguing opportunities for modulation of the microtubule cytoskeleton in the parasite life cycle and shows how material properties of microtubules in vitro can reflect important properties in vivo.

## Results

### Parasite-specific sequences modulate key tubulin ligand binding sites in *P. falciparum*

To gain insight into the architecture of the *P. falciparum* cytoskeleton, we first reconstituted and determined a high-resolution structure of microtubules using cryo-EM (Fig. 1a, Table 1). Microtubules were polymerized using tubulin isolated from *P. falciparum* and stabilized with paclitaxel[11]. To facilitate alignment of α- and β-tubulin during 3D reconstruction, microtubules were incubated with previously characterized recombinant *P. falciparum* kinesin 8B-motor domain prior to vitrification and imaging (Fig. 1a, blue density)[12]. Using a previously developed pipeline, we calculated the structure of *P. falciparum* 13-protofilament microtubules with an overall resolution of 3.2 Å (Supplementary Fig. 1a)[13,14]. Indeed, α- and β-tubulin could be distinguished clearly, as shown by their characteristic S9-S10 loops (Supplementary Fig. 1b). Since the microtubules were polymerized from single isoforms of α- and β-tubulin, the electron density was straightforwardly used for

automated model building and refinement to facilitate interpretation of this isoform-pure reconstruction (Supplementary Fig. 1c).

Tubulin is one of the most conserved eukaryotic proteins. However, *P. falciparum* tubulin is poorly conserved with respect to mammalian tubulin with α- and β-tubulin, exhibiting at most 85 % and 89 % identity, respectively (Supplementary Fig. 2a, b). Since most of our understanding comes from tubulin purified from either porcine or bovine brain, models arising from these studies will be referred to as brain tubulin and microtubules in the rest of the paper to simplify the comparisons. Our structure shows that the *P. falciparum* tubulins adopt a canonical fold (Fig. 1b). Consistently, superposition of the models of *P. falciparum* and brain tubulin monomers reveals small global Cα-backbone root mean square deviations (RMSDs) of 0.7 Å and 0.6 Å for α- and β-tubulin, respectively (Fig. 1c and Supplementary Fig. 2c, d). The largest difference between the tubulin sequences is found in the α-tubulin H1-S2 loop, which has a role in modulating lattice lateral contacts (Supplementary Fig. 2a, b). However, detailed analysis of this region in our structures is hindered by its characteristic flexibility meaning that the structural features could not be accurately modeled[15,16]. In contrast, a further set of non-conserved, well-modeled residues were scattered throughout each tubulin monomer such that their possible role in contributing to microtubule properties was not immediately obvious (Fig. 1d, shown in yellow). To begin to understand their possible effects on the parasite's cytoskeleton, we therefore examined more closely the configuration of the tubulin ligand-binding sites.

Hydrolysis of the β-tubulin-bound GTP at the longitudinal inter-dimer interface - the so-called exchangeable site (E-site) - causes changes in the microtubule lattice and drives polymer dynamics[4,17,18]. We therefore first explored sequence variations and structural differences between *P. falciparum* and brain tubulin at their GDP-bound E-sites. Strikingly, however, we observed no major differences in sequence or structure in the regions surrounding the β-tubulin-bound GDP (Fig. 1d, e). In contrast, although the non-exchangeable GTP binding pocket (N-site) of α-tubulin is itself very similar between the two tubulins, there are sequence variations extending for 2-3 residues at multiple positions in loops surrounding this site (Fig. 1d, Supplementary Figs. 2a, shown in orange and green). One notable difference lies at the N-terminus of the Gly-rich GTP binding loop, where three consecutive bulkier residues (His139, Ser140, Phe141) in brain tubulin are replaced by smaller amino acids (Ser139, Ala140, Val141) in *P. falciparum* (Supplementary Fig. 2a) and other apicomplexan parasites[19]. These differences affect the interaction of the Gly-rich loop with residues from the neighboring helix H5 and strands S5 and S6 (Fig. 1f). As a result, helix H5 in *P. falciparum* tubulin is kinked (RMSD of 1.0 to 2.2 Å for residues 193–195) towards the α-tubulin core, which is in turn accommodated by shifts (RMSD 0.3 to 1.6 Å) in the adjacent helices H11, H11′ and H12 that lie on the surface of α-tubulin (Fig. 1g). The precise positioning of these outer helices is potentially significant because they face the external side of the microtubule and serve as the binding site for multiple MAPs and motor proteins.

In addition to regulation by nucleotides, microtubule structure and dynamics can be modulated by numerous microtubule-targeting agents[20]. In our reconstruction, density corresponding to the microtubule-stabilizing drug paclitaxel is clearly visible in a well-characterized binding pocket in β-tubulin (Supplementary Fig. 1b). No major sequence or structural differences are detectable between *P. falciparum* and brain tubulin in this region, consistent with paclitaxel's generic binding (Fig. 1e and Supplementary Figs. 2b). Conversely, we used our structure to investigate the putative binding site of oryzalin, which binds to the dimer and preferentially inhibits parasite and plant microtubule polymerization[21–25]. Although its binding site has not been directly visualized, oryzalin is predicted to bind to α-tubulin in a cavity near the N-site, lined by strands S5 and S6[19,26–29] (Fig. 1g). Therefore, in addition to the previously proposed effect of sequence differences in

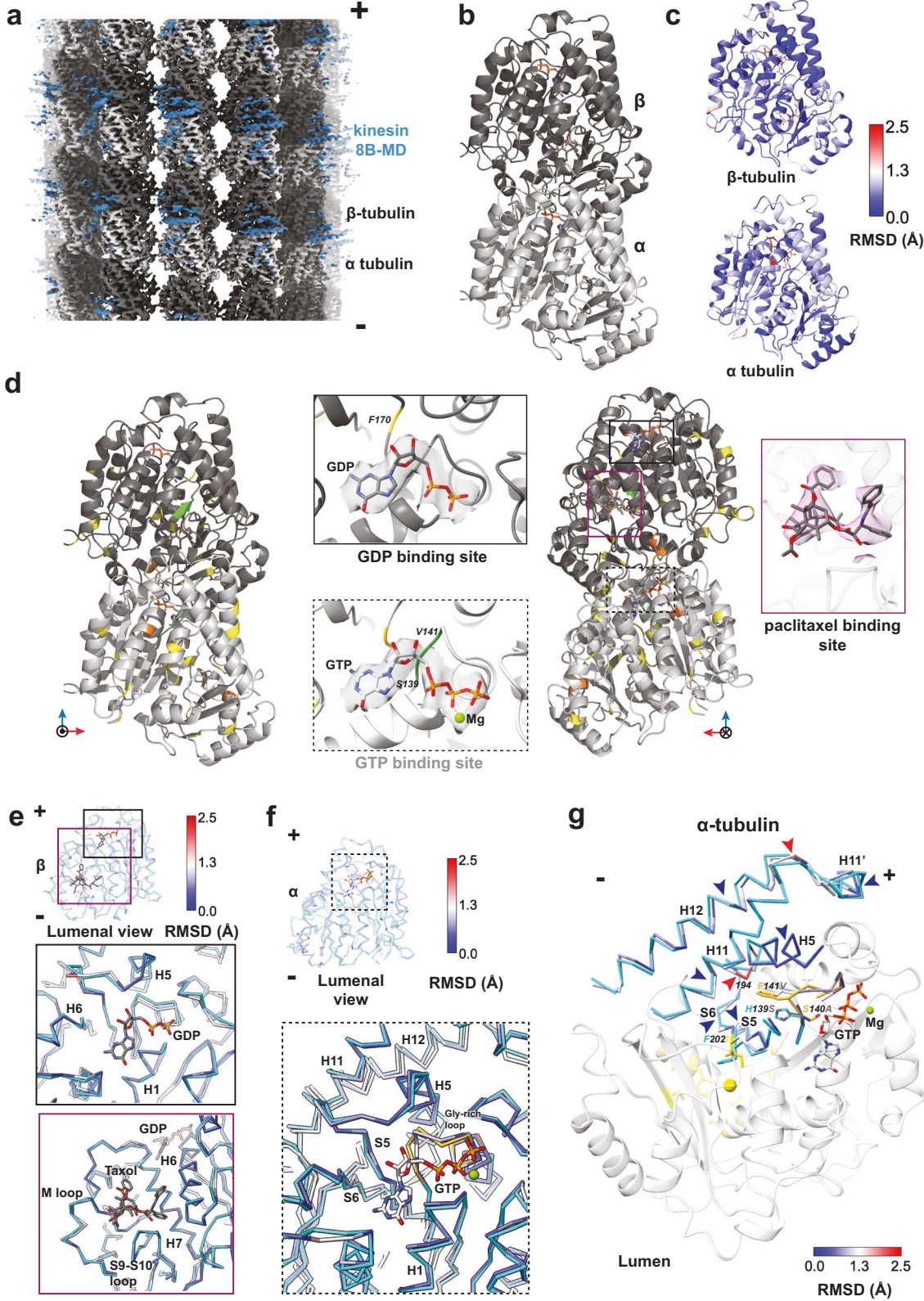

this region, our structure allows us to propose that displacements in these strands (RMSD 0.4 to 1.3 Å) also enhance the susceptibility of parasite tubulin to oryzalin inhibition compared to mammalian tubulin[11,19]. Taken together, our analyses suggest that although the overall structure of *P. falciparum* and brain tubulin is conserved, variations at key sites might account for differences in the properties of their microtubules.

## Cumulative effect of *P. falciparum* tubulin structural properties leads to distinct lattice interactions

While we only observe a few distinct characteristics in the *P. falciparum* tubulin dimer itself, microtubules are large polymers of tubulin in which these variations might have an additive effect. Therefore, we examined interactions between the tubulin dimers within the polymer lattice. To this end, we first superposed single protofilaments obtained

**Fig. 1 | Structure of *P. falciparum* tubulin reveals parasite-specific modulation of key ligand binding sites. a** Symmetrized reconstruction of *P. falciparum* 13-protofilament paclitaxel-stabilized microtubule, with density for α- and β-tubulin shown in light and dark gray respectively, and plus and minus ends indicated. Fragmented density corresponding to *P. falciparum* kinesin-8B motor domain (MD) used as a fiducial marker for alignment is shown in blue. **b** Model of *P. falciparum* tubulin dimer shown in cartoon depiction with bound ligands GTP and GDP shown in stick representation (α-tubulin: light gray; β-tubulin: dark gray). **c** Models of *P. falciparum* α- and β-tubulin monomers colored based on Cα -backbone RMSD obtained from structural superposition of *P. falciparum* and brain tubulins (porcine α1B- and β-tubulin; PDB ID:5SYF). **d** Model of *P. falciparum* tubulin dimer viewed from outside (left) and lumen (right) of the microtubule (α-tubulin: light gray; β-tubulin: dark gray), colored based on the number of consecutive non-identical residues between *P. falciparum* and brain tubulins (yellow- 1 residue; orange- 2 residues; green-3 or more residues) based on the alignment shown in Supplementary Fig. 2a and b. Insets show ligand binding sites for non-hydrolyzed GTP in α-tubulin, GDP, and paclitaxel on β-tubulin with segmented cryo-EM density map around the ligands. **e, f** Ligand binding sites in *P. falciparum* β-tubulin (**e**) and α-tubulin (**f**) superposed on respective brain tubulins (shown as Cα-backbone traces). The brain models are shown in cyan while *P. falciparum* models are colored based on RMSD. The Gly-rich loop is colored gold. The ligands GDP and paclitaxel (**e**) and GTP (**f**) are depicted as stick models. **g** The α-tubulin GTP binding loop with amino acid variations in *P. falciparum* (light gray cartoon) versus brain tubulin depicted in stick representation. Adjacent secondary structural elements (helices H5, H11, H12, strands S5 and S6), depicted as Cα-backbone traces, have been superimposed on the *P. falciparum* α-tubulin. The backbone trace of *P. falciparum* α-tubulin is colored based on RMSD obtained from comparison with brain tubulin (shown in cyan, Gly-rich loop in gold). Colored arrowheads indicate shifts in Cα- backbone of *P. falciparum*. Blue and red arrowheads depict RMSD ranges of 0.0 to 1.3 Å and 1.3 to 2.5 Å, respectively. The cavity beside the N-site, which is predicted to be the binding site for oryzalin, is indicated by a yellow sphere. The backbones of residues predicted to bind oryzalin are colored yellow.

by fitting atomic models of four tubulin dimers in the reconstructions of *P. falciparum* and brain (previously published) microtubules (Fig. 2a)[30]. The average tubulin intra- and inter-dimer distances are highly similar in both organisms (intra-dimer distance: *Pf*-40.7 Å, brain-40.9 Å; inter-dimer distance: *Pf*-41.1 Å, brain-40.9 Å). In addition, the longitudinal interfaces are also highly conserved (Supplementary Fig. 3a). However, as a result of the above described conformational shifts (Fig. 1g), we observed prominent twists in the C-terminal domain of α-tubulins. Although this twist does not affect the overall RMSDs of α-tubulins, we observed a greater shift at the interdimer interface (RMSD of C-domain 1.0 Å) than at the intradimer interface (RMSD of C-domain 0.7 Å) (Fig. 2a, blue box). The difference in the arrangement of tubulin sub-domains is propagated along the protofilaments where the topmost (4th) tubulin dimer most obviously shows these variations (Fig. 2a, red box). Thus, despite being the same length, the protofilament of *P. falciparum* microtubules exhibits a slight twist compared to brain microtubules (Supplementary Fig. 3b). Since this would be predicted to impact the inter-protofilament connectivity, we next explored the lateral interactions that form the cylindrical microtubule wall.

In neighboring α- as well as β-tubulins, lateral interactions are formed between the H1-S2, H2-S3 loops and H3 helix in one monomer and M-loops from the adjacent monomer. The conserved, key aromatic residue in the tubulin M-loops, His283 in α-tubulin and Tyr281 in β-tubulin, is buried in the interface (Fig. 2b). Our data show that in spite of sequence variations and a twist in the protofilaments of *P. falciparum* microtubules, the lateral interfaces are structurally comparable between the two species. Specifically, the conformations of the lateral loops are remarkably conserved at the α-α interface; this is despite amino acid differences over a five-residue stretch (41–45) in the disordered part of the H1-S2 loop, alongside a few variations in the H3 helix and the M-loop (Fig. 2c). At the β-β interface, the sequence and conformation of the M-loop are conserved in both species. Interestingly, while the conformation of H2-S3 loop is also conserved between the species, amino acid differences are observed between them (Fig. 2d and Supplementary Fig. 2b). In addition, there are residue variations in the H1-S2 loop which contribute to shifts in the loop and disordering of residues 37 to 39 (Supplementary Fig. 3c). Further, analysis of the residue composition of the H3 helix revealed more non-polar compared to polar residues at the β-β interface in *P. falciparum* microtubules (Fig. 2d; Supplementary Fig. 2a, b). While our reconstruction does not precisely visualize the interactions contributing to these changes, the cryo-EM density showed clear differences in side chain orientations between *P. falciparum* and brain microtubule lateral interfaces. As a result of these collective changes, both α-α and β-β lateral interfaces of *P. falciparum* microtubules exhibit a smaller buried surface area (calculated from PDBePISA[31]) than brain microtubules (Fig. 2c, d). Thus, a combination of distinctive properties of interface

residues with minor shifts in loops and side-chain positions allows the formation of a smaller lateral interface in *P. falciparum* microtubules. Since lateral and longitudinal interactions contribute to distinctive characteristics of lattices, we next examined the impact of changes in these interactions on the structural dynamics of *P. falciparum* microtubules.

## The *P. falciparum* microtubule lattice is stiffer than the brain microtubule lattice

The cryo-EM data we used to visualize *P. falciparum* microtubules provide precise yet static snapshots of lattice configurations. To analyze tubulin structural fluctuations, we therefore performed all-atom MD simulations on a lattice patch (3 protofilaments x 4 heterodimers) of *P. falciparum* GDP-paclitaxel microtubules and their brain counterparts (Supplementary Fig. 4a)[32]. Briefly, both *P. falciparum* and brain microtubule lattice patches were solvated in a cubic box, with their minus ends immobilized by applying position restraints to all heavy atoms of the three α-tubulins at the bottom layer. Three replicas of approximately 0.5 to 1 μs runs were performed for each structure, and equilibrium trajectories were collected for further analysis after the RMSDs had stabilized (Supplementary Fig. 4b). By the end of the simulations, both *P. falciparum* and brain microtubule lattices exhibited outward radial bending and clockwise twisting of protofilaments at the plus end[33,34]. Interestingly, the *P. falciparum* lattice patch appeared to be overall stiffer than the brain one (Fig. 3a).

To examine how the behavior of these lattice patches is defined by the properties of each tubulin dimer, its physical characteristics were extracted. We used a combination of correlated collective variables (CVs) along the different axes of movement, called tangential swing and twist bending, to analyze the bending-torsional dynamics of the tubulin dimers in the lattice patches[33] (Fig. 3b, c) The *P. falciparum* patch demonstrated significantly lower tangential swing as well as twist-bending (15.9° ± 1.0° and 10.0° ± 1.9° respectively) compared to its brain counterpart (25.3° ± 0.6° and 18.7° ± 0.6° respectively) (Fig. 3b, c), underscoring its lower overall flexibility.

To further explore the basis for these differences in the flexibility of *P. falciparum* and brain microtubule lattice patches, we employed a coarse-grained (CG) approach with a heterogeneous elastic network model (HeteroENM)[35,36]. Here, each tubulin was coarse-grained into a single CG bead positioned at the centroid of each monomer (Fig. 3d). The connectivities between each monomer were represented by interlinked harmonic springs and the dynamic lattice interactions between them were parameterized using HeteroENM. These analyses can dynamically explore the strength of the lattice contacts. The experiments showed that while longitudinal interdimer interactions were similar between the two tubulins (Fig. 3e p-value = 0.21 > 0.05), the lateral interfaces (of which there are three per monomer, Fig. 3d) of *P. falciparum* were collectively significantly stronger than the brain

**Table 1 | Cryo-EM data collection, refinement and validation statistics**

| | 13-protofilament *P. falciparum* paclitaxel-stabilized GDP micro-tubule (EMD-53571, https://www.ebi.ac.uk/pdbe/entry/emdb/EMD-53571) (PDB ID: 9R4X) | 15-protofilament *P. falciparum* GMPCPP microtubule (EMD-53572, https://www.ebi.ac.uk/pdbe/entry/emdb/EMD-53572) (PDB ID: 9R4Y) |
|---|---|---|
| **Data collection and processing** | | |
| Magnification | 81000x | 81000x |
| Voltage (kV) | 300 | 300 |
| Electron exposure (e–/Å²) | 50.75 | 48.51 |
| Defocus range (µm) | −0.6 to −2.4 | −0.7 to −2.2 |
| Pixel size (Å) | 1.067 | 1.067 |
| Symmetry imposed | Helical | Helical |
| Micrographs (no.) | 4562 | 3854 |
| Initial particle images (no.) | 140038 | 115668 |
| Final particle images (no.) | 30442 | 41886 |
| Map resolution (Å) | 3.2 | 2.9 |
| FSC = 0.143 | | |
| Map resolution range (Å) | 3.1–4.8 | 2.8–3.9 |
| **Refinement** | | |
| Initial model used (PDB code) | An initial model obtained in ModelAngelo using the protein sequence of *P. falciparum* tubulin | The refined atomic model for a tubulin dimer from the 13-protofilament *P. falciparum* GDP-paclitaxel microtubule was used as an initial model |
| Model resolution (Å) | 3.1 | 2.8 |
| FSC = 0.143 | | |
| Map sharpening *B* factor (Å²) | -50 | -50 |
| Model compositions | | |
| Non-hydrogen atoms | 6762 | 6726 |
| Protein residues | 850 | 853 |
| Ligand | 4 | 4 |
| *B* factors (Å²) | | |
| Protein | 102.67 | 111.60 |
| Ligand | 109.89 | 96.49 |
| R.m.s. deviations | | |
| Bond lengths (Å) | 0.004 | 0.004 |
| Bond angles (°) | 0.928 | 0.937 |
| Validation | | |
| MolProbity score | 1.63 | 1.51 |
| Clashscore | 7.00 | 5.68 |
| Poor rotamers (%) | 1.66 | 1.38 |
| Ramachandran plot | | |
| Favored (%) | 97.62 | 97.52 |
| Allowed (%) | 2.38 | 2.48 |
| Disallowed (%) | 0.00 | 0.00 |

microtubule contacts (Fig. 3f, p-value = 0.025 < 0.05). Taken together, this can explain the smaller bending-torsional dynamics observed in *P. falciparum* microtubules.

So far, our analysis provides molecular perspectives on the *P. falciparum* lattice from cryo-EM, while the MD calculations provide insights for understanding the flexibility of partially restrained tubulin patches. To further evaluate protofilament flexibility, we used cryo-electron tomography (cryo-ET) to compare the plus ends of *P. falciparum* and brain microtubules polymerized in vitro and stabilized by paclitaxel (Fig. 3g, Supplementary Fig. 5a, b). We define these ends as the region beyond which the microtubule is no longer a complete cylinder. The microtubule plus ends in both populations revealed a range of lengths and curvatures of bent protofilaments (Fig. 3h). The

curvature observed in the presence of paclitaxel is lower than that seen for dynamic microtubules, consistent with the previously described effect of paclitaxel in flattening the protofilament curls[37–39]. While the curvature of the protofilaments was similar in both species, *P. falciparum* microtubules showed substantially shorter individual protofilaments at the plus end (Fig. 3i, j).

How the stiffness of an individual tubulin dimer affects the stiffness of a protofilament and then a whole microtubule continues to be an area of active research (see Discussion)[33,37,38,40–43]. The results of our cryo-ET and MD experiments conducted to investigate the properties of *P. falciparum* tubulin may initially appear paradoxical, but they reveal different facets and scales of its behavior. In fact, results from MD simulations in our study, including both the stiffness of the tubulin

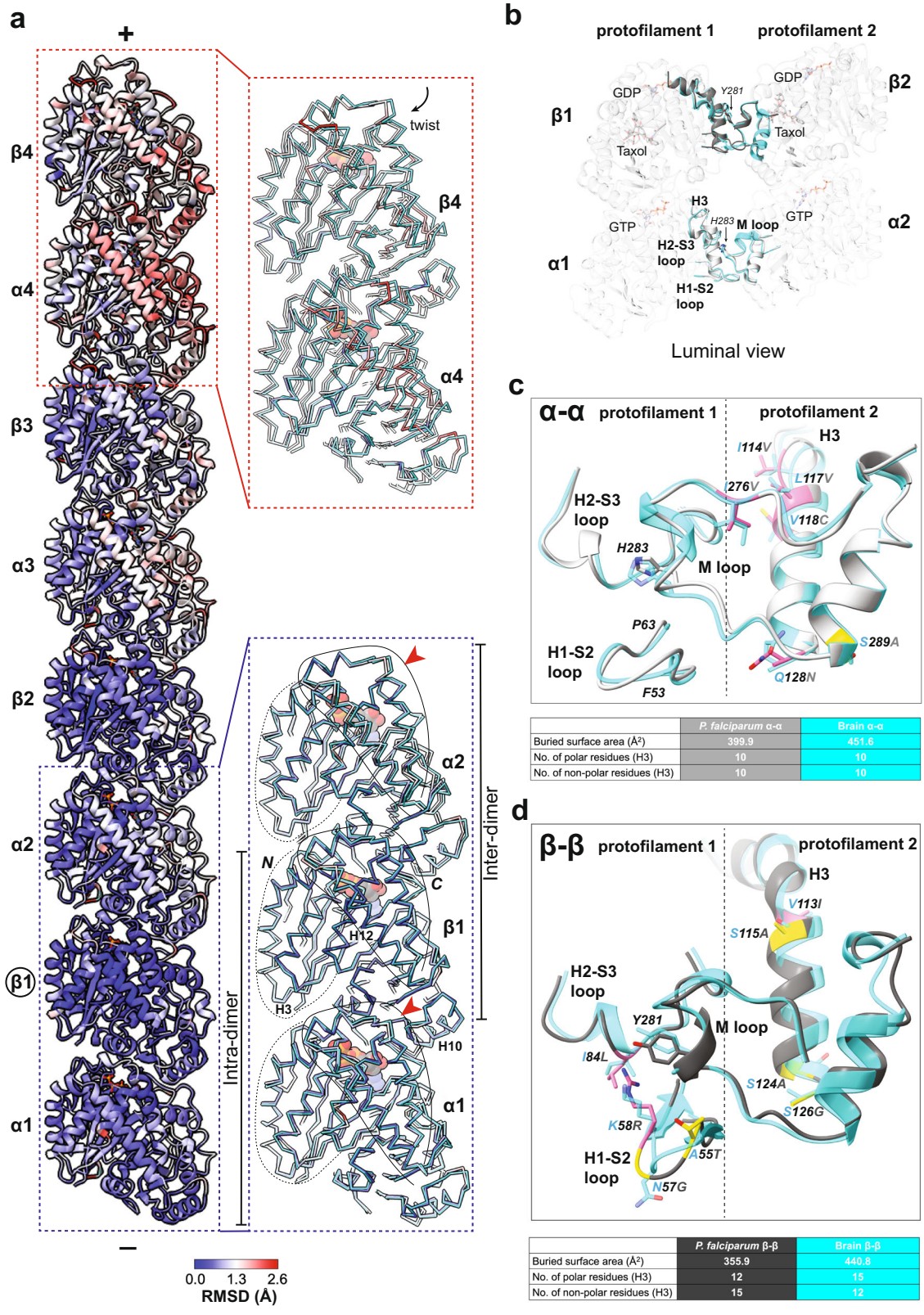

by stronger lateral contacts producing shorter end structures[34]. In addition, the lateral connections are more readily maintained by the restricted movement in their stiffer protofilaments[33,40]. The shorter lengths of *P. falciparum* microtubule plus ends obtained from cryo-ET can therefore be attributed to a combination of stronger lateral interactions and lower protofilament flexibility of the *P. falciparum* microtubule lattice. Thus, the distinctive properties of *P. falciparum*

dimer and strength of the lateral contacts in the lattice patch, can help us understand the behavior of the individual *P. falciparum* protofilaments observed in our cryo-ET experiments (Fig. 3g–i). Specifically, *P. falciparum* microtubules show short plus ends, whereas single and clusters of long, splayed protofilaments are observed in brain microtubules (median = 14.8 nm compared to 21.3 nm). This is consistent with the previously described behavior of protofilaments connected

**Fig. 2 | Structural properties of the *P. falciparum* tubulin dimer produce distinct lattice interactions. a** A four tubulin dimer protofilament from a *P. falciparum* 13-protofilament microtubule colored by RMSD from superposition on β1-tubulin (circled) of a brain microtubule protofilament model (porcine α1B- and β-tubulin; PDB ID:5SYF). Plus and minus ends of the protofilament are indicated. Zoomed-in views of the outlined regions are depicted in insets showing Cα-backbone traces of indicated regions colored in cyan for the brain model and by RMSD for *P. falciparum,* respectively. N- and C-terminal domains of tubulin have been outlined, and nucleotides are represented as space-filling spheres. Shifts in the C-terminal domain are indicated by red arrowheads. **b** Overview of the lateral interfaces between α- and β-tubulin from *P. falciparum* and brain microtubules, with key ligand binding sites indicated (*P. falciparum* α-tubulin: light gray; *P. falciparum* β-tubulin: dark gray; brain α- and β- tubulin: cyan). Interacting loops and key residues from neighboring tubulins are indicated and shown in cartoon representation. **c, d** Comparison of *P. falciparum* and brain microtubule lateral interactions. The interacting loops are depicted as cartoon Cα backbone models and colored light gray for *P. falciparum* α-tubulin, dark gray for *P. falciparum* β-tubulin and cyan for brain α- and β- tubulin. Side chains of residues that differ are indicated. *P. falciparum* residues are colored based on the sequence conservation scheme outlined in Supplementary Fig. 2a, b (pink- similar residues; yellow- 1 non-identical residue; orange- 2 non-identical residues; green-3 or more non-identical residues), with quantification of the interface properties tabulated below.

tubulin that lead to variations in lattice interactions also reveal a distinct mechanism for regulating physical properties of the polymer ends, potentially influencing microtubule mechanics.

## E-site nucleotide determines the microtubule architecture in *P. falciparum* but does not alter stiffness

Our analyses of 13-protofilament *P. falciparum* microtubules showed distinct properties resulting from the parasite tubulin's distinct behavior. However, in different stages of the *P. falciparum* life cycle, microtubules show a range of 13- to 18-protofilament architectures. To increase the variety of protofilament architectures in our experiment, we used GMPCPP, a non-hydrolysable analogue of GTP which is known to influence microtubule lattice properties[16,17,44,45].

First, using total internal reflection fluorescence (TIRF) microscopy, we observed that *P. falciparum* tubulin formed long microtubules when stabilized by paclitaxel (Fig. 4a). In contrast, we observed small clusters of tubulin with GMPCPP (Fig. 4a). We also examined such tubulin clusters by negative stain EM and observed very short GMPCPP-stabilized microtubules (Fig. 4b). We then used cryo-EM (Supplementary Fig. 6a) to compare the variation in architectures of *P. falciparum* microtubules based on the nucleotide used in polymerization (Fig. 4c). While *P. falciparum* GDP-paclitaxel microtubules assembled into a majority of 13-3 (25%) and 14-3 (35%) architectures, 60% of GMPCPP microtubules exhibited 15-3 protofilament architecture. In contrast, and consistent with previous reports, the protofilament distribution in GMPCPP-microtubules assembled from TOG-purified (see Methods) brain tubulin included 13- and 14-protofilament microtubules (Fig. 4c)[16,18,45,46]. This suggests that the tendency to form 15-protofilament GMPCPP microtubules is an intrinsic and distinctive property of *P. falciparum* tubulin.

To better understand the 15-3 protofilament architecture of *P. falciparum* GMPCPP microtubules, we determined the 3D structure of this polymeric form (Fig. 4d, Supplementary Fig. 6a and Table 1). α- and β-tubulin could clearly be distinguished and modeled in the 2.9 Å symmetrized 3D reconstruction (Supplementary Fig. 6b, c). While the individual α- and β-tubulin models from GMPCPP and GDP-paclitaxel reconstructions are nearly identical, the dimer spacing within the GMPCPP protofilaments corresponds to an extended tubulin conformation (Intradimer distances: 40.5 Å-GDP-paclitaxel, 40.9 Å-GMPCPP; Interdimer distances: 41.4 Å-GDP-paclitaxel, 42.5 Å-GMPCPP), as has been previously reported for microtubules with GTP-like analogues bound to the E-site[16,47–49]. We also conducted the same MD simulations as described above on GMPCPP lattice patches (Fig. 4e and Supplementary Fig. 6d). We observed similar overall behavior for protofilament twist-bending and tangential swing variables - that is, the *P. falciparum* tubulin is stiffer than brain tubulin. However, when comparing within the same organism, the torsional dynamics of the *P. falciparum* tubulin are similar in GMPCPP and GDP-paclitaxel lattice patches. In contrast but consistent with other reports, the brain GMPCPP lattice patches are stiffer than the GDP-paclitaxel patches (Figs. 3b, 3c and 4e)[40,50]. Overall, these findings indicate that independent of the E-site nucleotide, the intrinsic properties of *P.* *falciparum* tubulin and the lattice contacts it forms alter the properties of the protofilaments and microtubules.

## In vitro reconstituted microtubules show high structural similarity with in vivo microtubule architectures

The range of microtubule architectures formed in vitro by *P. falciparum* tubulin is intriguing because, in the gametocyte stage of its life cycle, *P. falciparum* assembles large microtubules, with 15- and 17-protofilament architectures being the most prominent in nuclear and subpellicular microtubules, respectively[9]. This is noteworthy as these architectures are distinct from the canonical 13-protofilament architecture found in most eukaryotic cells[8]. To understand how the *P. falciparum* microtubule can accommodate two or more additional protofilaments, we first compared the in vitro assembled 15-protofilament (GMPCPP) and 13-protofilament (GDP paclitaxel) reconstructions. The 15-protofilament lattice carries a protofilament skew of 1.1° (Supplementary Fig. 7a), consistent with the long-standing lattice accommodation model that predicts that protofilament skew is required to accommodate protofilaments greater than 13[51]. Moreover, we observed that *P. falciparum* 15-protofilament (GMPCPP) microtubules are close to perfectly cylindrical, while 13-protofilament (GDP-paclitaxel) microtubules are less regular, as evidenced by structural variations near the seam (Fig. 4f). Although it remains unclear whether this arises from specific properties of the tubulin, the mode of stabilization or the structure, cylindrical architectures that demonstrate lattice homogeneity could contribute to microtubule stability in their physiological context[16,30].

To further understand the consequences of integrating additional protofilaments on the *P. falciparum* polymer lattice, we examined the inter-protofilament lateral contacts. Both α-α and β-β interfaces were similar in 13- and 15-protofilament microtubules (Fig. 4g, table). In contrast, we observed smaller surface areas for lateral contacts of *P. falciparum* microtubules in an equivalent comparison with brain microtubule structures (Fig. 4g), despite the remarkable structural conservation of the loop backbones. This further reinforces the earlier observation that while the E-site nucleotide affects the protofilament distribution of *P. falciparum* microtubules, it does not substantially modify the tubulin interactions within the lattice.

How do these in vitro microtubule structures compare to what is seen in the parasite itself? We compared the structures of the 15-protofilament microtubules polymerized in vitro and microtubules in vivo (from gametocytes) obtained using subvolume averaging (Fig. 5a). Both microtubule reconstructions have comparable dimensions and the in vitro structure fits well into the in vivo structure (Fig. 5b). As expected, tubulin dimers from the high-resolution in vitro 3D reconstruction also accurately map to corresponding positions in the subvolume average (Supplementary Fig. 7b). While we could not examine intricate details in the in vivo microtubules due to the limited resolution of the subvolume average, the close structural similarity of our in vitro and in vivo reconstructions allowed us to make further refinements of the previously published in vivo structure[9]. This revealed the previously undetectable protofilament skew in the in vivo polymers (Supplementary Fig. 7c, EMD-15536), and the high similarity allows us to

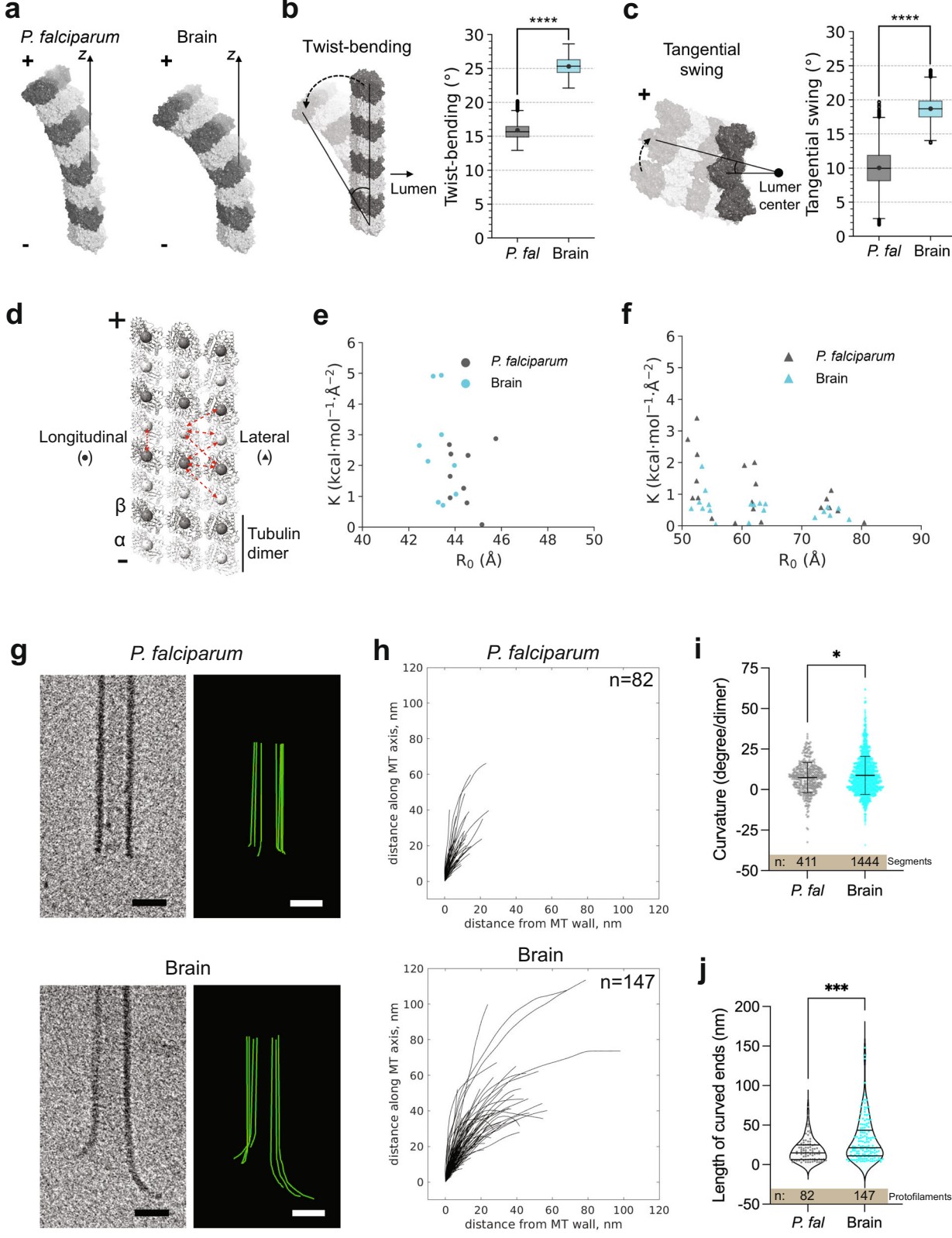

propose that the polymer stiffness we characterized in vitro reflects the properties of the polymer within the parasite. While microtubules certainly contribute to parasite mechanics, their quantitative contribution to overall rigidity remains to be determined[52,53]. However, stiffer microtubules could have the physiological consequence of making the parasite less pliable before maturation, preventing early release from sites of sequestration during gametocyte development[9,54,55].

## Discussion

The contrast between the stringency with which in vivo microtubule architecture is defined, albeit with extraordinary diversity across organisms[8] and the architectural variability of microtubules polymerized in vitro has proved a source of fascination and curiosity for decades[56]. The multiple, dynamic lattice contacts that define microtubule nucleation and growth are influenced in vitro in only partially

**Fig. 3 | Structural and dynamic properties of *P. falciparum* and brain microtubules reveal differences in protofilament flexibility. a** Representative configurations of *P. falciparum* and brain GDP-paclitaxel microtubule protofilaments (porcine α1B- and β-tubulin; PDB ID:5SYF) derived from molecular dynamics (MD) simulations, highlighting the outward bending and clockwise twisting of plus end protofilaments. **b** Left: Twist-bending angle, characterized by angular deformation of protofilaments. The solid lines show the reference straight axis and the bent protofilament axis defining the twist-bending angle indicated by a dashed arrow. Right: Twist bending dynamics for a tubulin dimer obtained from *P. falciparum* (*P. fal*) and brain microtubule lattice patches. The box represents the interquartile range (IQR), extending from the first quartile (Q1) to the third quartile (Q3), with a horizontal line indicating the median. Whiskers extend from the edges of the box to the most extreme data points within 1.5×IQR from the box. Data points beyond the whiskers, referred to as outliers, are shown individually. The solid dot within each box indicates the mean value of twist-bending angles. $n = 5$, using block averaging, the trajectory was cut into 5 pieces, and average was calculated for each piece. The 5 averaged data points were used for statistical analysis using the two-sided t-test. ****, $p = 2.2*10e-07$ **c** Left: Tangential swing angle measured along the protofilament axis. The straight lines indicate the lateral reference plane and the central protofilament axis that form the tangential swing angle indicated by a dashed arrow. Right: Tangential swing dynamics for a tubulin dimer (*P. falciparum, P. fal;* brain). Box plots follow the same labeling scheme as in panel (b), with medians, IQRs, whiskers, outliers, and mean values explicitly shown. Statistical analysis was done in the same manner as above. ****, $p = 2.7*10e-05$ **d** Heterogeneous elastic network model (HeteroENM) analysis was carried out on a lattice patch of the microtubule shown in cartoon representation (α-tubulin: light gray; β-tubulin: dark gray). A single bead coarse-grained at the centroid of each tubulin monomer is depicted as a sphere. The longitudinal and lateral interactions analysed are indicated by red double-headed arrows. **e** HeteroENM analysis of longitudinal interactions, with spring constants (K) are plotted against equilibrium bond lengths ($R_0$) for *P. falciparum* and brain microtubule longitudinal contacts. Two-sided t-test was used for statistical analysis, $n = 9$. **f** HeteroENM analysis of lateral interactions for *P. falciparum* and brain microtubules, as in d. Statistical analysis was done by two-sided t-test, $n = 20$ **g** Tomogram slices (left) and protofilament traces (right) of *P. falciparum* (top) and brain (bottom) paclitaxel-stabilized microtubule (bovine tubulin) ends. Scale bars = 20 nm. **h** Plus end protofilament traces aligned at their origin from *P. falciparum* (top) and brain (bottom) microtubules. 18 *P. falciparum* and 26 brain microtubules were analysed. Distribution of **i** curvatures and **j** lengths of protofilament curls at plus ends of *P. falciparum* and brain microtubules. Curvatures calculated per curved segment of the microtubule plus end are plotted as circles and the mean ± SD is indicated. Statistical analysis was carried out using the two-tailed t-test with Welch's correction. *, $p = 0.02$. Lengths measured per protofilament traced at the plus end are plotted as single points. The median and interquartile range are indicated by the horizontal lines in the violin plot. Two-tailed Mann-Whitney test was used for statistical analysis. ***, $p = 0.0002$.

predictable ways, although accumulating data show that buffer conditions, temperature, nucleotide state, and tubulin sequence can all influence the architectural variability of the resulting microtubule population[16,17,44,45,49,57–63]. In this study, we used natively purified, single isoform, minimally post-translationally modified *P. falciparum* tubulin, enabling us to be confident that all the lattice contacts that contribute to the microtubules we imaged are chemically identical[11]. We nevertheless observed a range of microtubule architectures under all experimental conditions, highlighting the complexity of the processes involved. We found that while *P. falciparum* tubulin can form canonical 13-protofilament microtubules in the presence of GTP and stabilized by paclitaxel, GMPCPP microtubules are unexpectedly dominated by non-canonical 15-3 microtubules, one of the least reported architectures in structural databases. Within the *P. falciparum* parasite, PTMs, MAPs, and other differentially expressed nucleating factors may account for some of the architectural variation in different life stages. Crucially, our work also captures the specific properties of *P. falciparum* tubulin that contribute to the architectural variability of these in vitro polymerized microtubules. Subtle variations in α-tubulin sequence and three-dimensional structure produce a slight protofilament twist within polymers, stiffen lattice patches and reduce the lengths of curved protofilaments at microtubule plus ends, all of which influence the energetics of lattice bond formation within the polymers (Fig. 5c). Protofilament variations in the parasite could also be influenced by the α2-tubulin isoform, the only other isoform of α- and β-tubulins, the sequence of which varies from the α1- isoform mostly in the C-terminal tails and H1-S2 loop regions. These regions are disordered in most microtubule structures determined by cryo-EM, including ours. As this study shows, sequence information alone is insufficient to predict structural variations between isoforms, and functional analyses will be necessary to determine the impact of amino acid differences in α2-tubulin. Our findings add to the growing body of observations on the in vitro polymerization behavior of tubulins from different sources and, importantly, also describe intrinsic properties of *P. falciparum* tubulin that may contribute to architectural variability within the parasite.

The critical involvement of microtubules in many cellular processes, together with the numerous small molecule binding sites on tubulin, have made microtubules an attractive therapeutic target, including in the context of infectious diseases[20]. As a result, there are a number of extensively utilized microtubule targeting agents that are effective in a range of organisms[64]. However, such pleiotropic

compounds are not therapeutically useful in the context of infectious diseases, where compounds that selectively target the parasite but not the host cytoskeletons are critical. This highlights that the precise molecular basis for such selectivity has been less well understood and remains unpredictable. This likely reflects the small sequence differences in tubulin from different species, but perhaps more importantly, the allosteric complexity of tubulin. For example, microtubules from many organisms, including *P. falciparum* are stabilized by paclitaxel binding to β-tubulin[30,65–67]. Indeed, we visualized paclitaxel bound to β-tubulin in our *P. falciparum* 13-protofilament reconstruction (Fig. 1), implying that paclitaxel binding site ligands would be poor targets for intracellular parasite-specific treatments. In contrast, the selective inhibitory properties of oryzalin and other dinitroaniline compounds towards polymerization of plant and protozoan (including *Plasmodium*) microtubules have been known for some time[68]. Although dinitroaniline binding to tubulin has yet to be directly visualized, our structure suggests that selectivity of these compounds might arise both from the previously identified sequence variations around its predicted binding site and from shifts in adjacent regions of α-tubulin. Despite the wealth of available structures of brain tubulin, we need tubulin and microtubule structures from therapeutically important sources to understand how minor differences can predict small molecule sensitivity. Indeed, the use of protozoan tubulin as an experimental model recently enabled the targeted design of an antiparasite small molecule[69]. As structures from more diverse tubulin sources are determined, including visualization of unpolymerized tubulin dimers by cryo-EM, the subtle origins of susceptibility to small molecule binding will become more predictable, enabling more rational design of precise microtubule perturbing drugs.

The mechanistic relationship between loop structuring, lateral interaction strength, strain accumulation, and the resulting mechanical properties of microtubules, which are typically in the micrometer range, remains intricate and difficult to predict. An example of this is the increased strength of lateral interactions observed in *P. falciparum* lattice patches, despite their reduced buried surface area compared with the brain microtubule lattice. A possible explanation for this could be changes in the composition of interface residues leading to lateral interactions with varying strengths. However, given that the *P. falciparum* tubulin dimers polymerize with similar longitudinal interactions but stronger lateral contacts, the expectation would be that their polymerization dynamics are broadly similar to brain tubulin but exhibit different mechanical properties. Indeed, *P. falciparum*

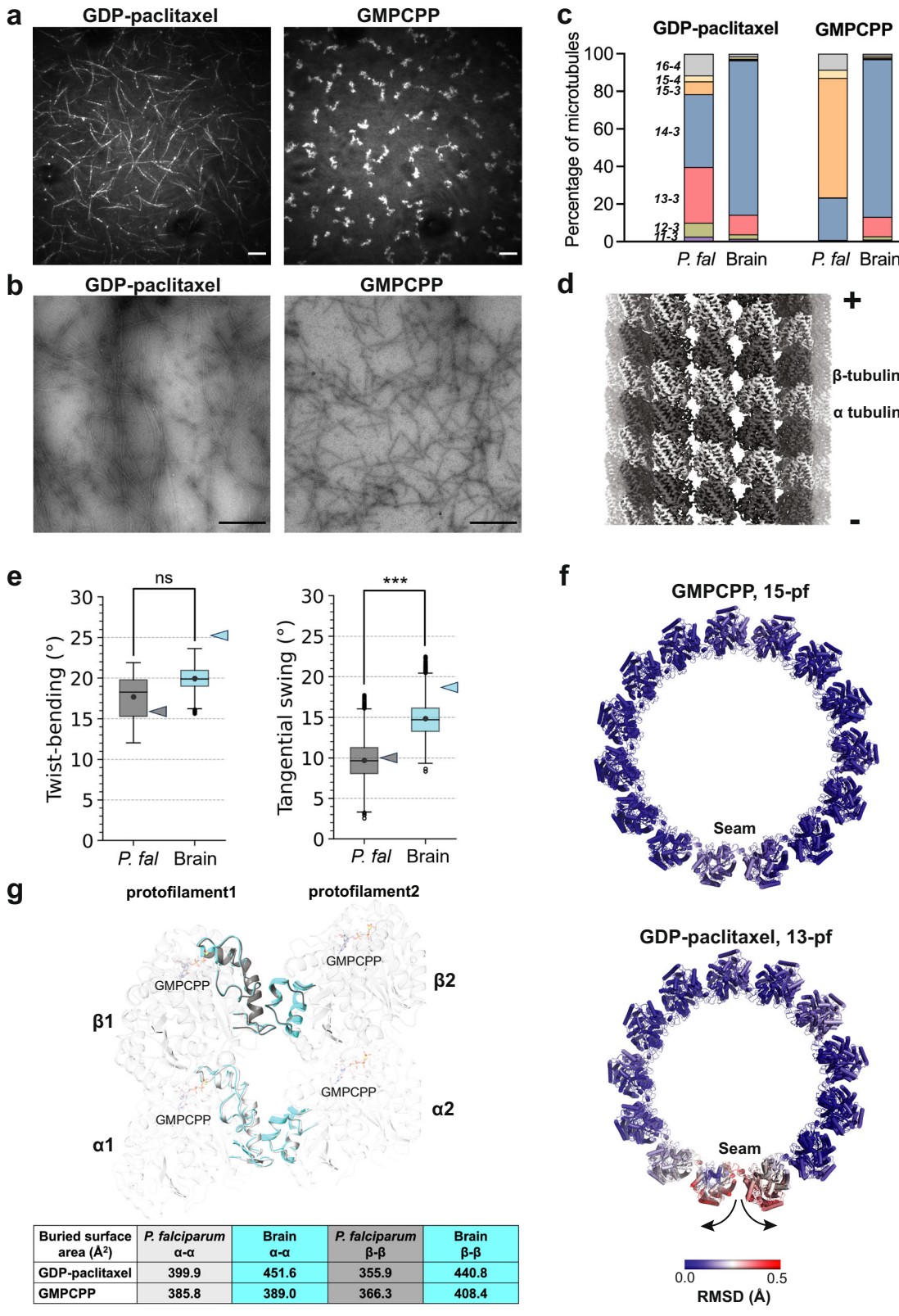

microtubules share a number of dynamic properties with brain microtubules[11]. Further, our MD and cryo-ET data highlight different biophysical properties of the microtubule ends and, by inference, of the polymers themselves. Not surprisingly, the microtubule plus end, characterized by its transient and heterogeneous shapes, will likely exhibit distinct mechanical properties compared to the microtubule lattice. Future developments in the field will help to extend the exact findings from a lattice segment to the entire polymer. Crucially, however, our study demonstrates the intrinsic tendency of purified parasite tubulin to produce parasite-relevant architectural variants in vitro. A key additional insight from in vivo characterization of *Plasmodium* microtubules is the presence of distinct repertoires of MAPs that likely play critical roles in further tuning the properties of particular populations of cytoskeleton filaments, for example sub-pellicular compared

**Fig. 4 | *P. falciparum* tubulin forms 15-protofilament microtubules with GMPCPP. a** Representative TIRF microscopy images of GDP-paclitaxel (left) and GMPCPP (right) microtubules polymerized at 37 °C for 20 minutes in the presence of 10% fluorescently labeled brain tubulin for visualization. Scale bars = 10 μm. **b** Representative negative stain micrographs of *P. falciparum* GDP-paclitaxel (left) and GMPCPP (right) microtubules polymerized using 4 μM *P. falciparum* tubulin only, imaged at 4400x. Scale bars represent 2 μm. **c** Distribution of protofilament architectures for *P. falciparum* (*P. fal*) and brain (bovine) GDP-paclitaxel and GMPCPP microtubules. **d** Symmetrized reconstruction of *P. falciparum* 15-3 protofilament GMPCPP microtubule (α-tubulin: light gray; β-tubulin: dark gray), with minus and plus ends indicated. **e** Twist-bending and tangential swing dynamics calculated for tubulin dimers from the GMPCPP microtubule lattice. The box represents the interquartile range (IQR), extending from the first quartile (Q1) to the third quartile (Q3), with a horizontal line indicating the median. Whiskers extend from the edges of the box to the most extreme data points within 1.5×IQR from the box. Data points beyond the whiskers, referred to as outliers, are shown individually. $n = 5$, the trajectory was divided into 5 pieces using block averaging and an average was calculated for each piece. The 5 averaged data points were used in the statistical two-sided t-test. ns, $p = 0.096$; ***, $p = 0.0014$. Corresponding values for GDP paclitaxel lattices are indicated in the plot as colored triangles. **f** Minus end view of one helical turn of a *P. falciparum* microtubule colored based on Cα RMSD between the models from C1 and symmetrized reconstructions of GMPCPP (left) and GDP-paclitaxel (right) microtubules. **g** Superposition of respective α-α and β-β lateral interfaces from *P. falciparum* and brain (porcine α1B- and β-tubulin; PDB ID: 6DPU) GMPCPP microtubules (*P. falciparum* α-tubulin: light gray; *P. falciparum* β-tubulin: dark gray; brain α- and β- tubulin: cyan). Interacting loops from neighboring tubulins are shown in cartoon representation, with quantification of the interface properties tabulated below.

## Methods

### Ethics statement

Human red blood cell (RBC) concentrates were obtained from the DRK-Blutspendedienst Nord-Ost (German Red Cross Blood Donation Service North-East). All samples were fully anonymized prior to receipt, and the supplier provided a formal confirmation of anonymization. All donor units were screened and tested negative for HIV, HBV, and HCV. In accordance with local regulations, research involving *Plasmodium* and human blood was conducted under the Official Authorization from the Landesamt für Gesundheit und Soziales (LAGeSo) Berlin, reference number IV C 26 - 202/01-24. Bovine brain tissue for tubulin purification was obtained from the Bundesinstitut für Risikobewertung as a byproduct. No animals were sacrificed specifically for this research; therefore, specific animal ethics committee approval was not required.

### Cultivation of *P. falciparum* blood stages

Asexual *P. falciparum* parasites were cultured as previously described[71]. Briefly, *P. falciparum* 3D7 was maintained at 3% hematocrit in O+ erythrocytes in RPMI-1640 medium supplemented with 25 mM HEPES, 0.2% (w/v) sodium bicarbonate, 200 μM hypoxanthine, 10 mM D-glucose, 20 μg/ml gentamycin, and 0.5% Albumax II. Parasites were cultured on a horizontal shaker in a 5% $CO_2$, 5% $O_2$, and 90% $N_2$ atmosphere at 37 °C. The media was changed daily by centrifuging at $500 \times g$ for 5 minutes and replacing the supernatant with fresh media. Cultures were synchronized by sorbitol hemolysis, achieved by incubating the cells in 5% (w/v) sorbitol for 10 minutes at 37 °C. Parasitemia was checked every 48 h by Giemsa Smear and maintained at 1–10% by diluting the cultures and replenishing the hematocrit with uninfected erythrocytes.

### Purification of *P. falciparum* and bovine brain tubulin

*P. falciparum* and bovine brain tubulin were purified as previously described[11]. Briefly, *P. falciparum* parasites were isolated from infected erythrocytes by saponin lysis. The infected erythrocytes were resuspended in 0.15% (w/v) saponin in 1x PBS and incubated on ice for 15 min, with occasional tube inversions. The isolated parasites were washed thrice by centrifugation for 10 min at 2000 x $g$, 4°C, and resuspension in ice-cold PBS. Finally, the parasite pellet was flash-frozen in liquid nitrogen and stored at −80°C. For *P. falciparum* tubulin purification, parasite aliquots were quickly thawed and then resuspended in a minimal volume of 1x BRB80 supplemented with 100 μM GTP, 2 mM DTT, and 1x protease inhibitors on ice. For bovine brain tubulin purification, brain tissue stored at −80°C was pulverized using a mortar and pestle on dry ice and resuspended in 1x BRB80 supplemented with 100 μM GTP, 2 mM DTT, 1x protease inhibitors, 30 U/ml benzonase, and 10 μg/mL Cytochalasin D on ice.

Tubulin purification then proceeded by sonicating the cell suspension at full power in 0.5 seconds intervals for a total of 30 seconds using a Sonoplus HD 2070 sonicator equipped with a MS73 probe, followed by centrifugation in an MLA-80 rotor for 10 minutes at 440,000 x $g$ at 4°C. The supernatant was loaded at 0.5 column volume (CV)/min onto a TOG-column preequilibrated with BRB80. The flow rate was then changed to 1 CV/min for the following wash steps: 1) 10 CV of 1x BRB80, 100 mM GTP; 2) 3 CV of 1x BRB80, 100 mM GTP, 10 mM $MgCl_2$, 5 mM ATP followed by a 15-min incubation; 3) 5 CV of 1xBRB80 and 100 mM GTP. The tubulin was eluted with 3 CV of 1x BRB80, 100 mM GTP, and 500 mM $(NH_4)_2SO_4$. Pooled fractions were desalted into 1x BRB80 and 10 mM GTP with a PD10 desalting column and concentrated using an Amicon Ultra 30 K MWCO centrifugal filter. Tubulin was aliquoted and snap frozen in liquid nitrogen.

### Negative stain sample preparation and electron microscopy

GDP-paclitaxel microtubules were polymerized by incubating *P. falciparum* tubulin (5 μM) with 2 mM GTP and 5 μM paclitaxel (paclitaxel, Sigma-Aldrich Cat No. 580555, dissolved in DMSO) in BRB80 buffer (80 mM PIPES, 2 mM $MgCl_2$, 1 mM EGTA, pH 6.8) on ice for 10 min and in a water bath maintained at 37 °C for 1 h. For GMPCPP microtubules, *P. falciparum* tubulin (4 μM) was incubated with 1 mM GMPCPP in BRB80 buffer on ice for 10 min and then 37 °C for 1 h. 3 μL of each of the samples were added to glow-discharged continuous carbon film coated 400 mesh copper grids (Pacific Grid Tech). After 30 seconds, staining solution (2% uranyl acetate in water) was added drop by drop to the grid. Excess stain was blotted off after 30 seconds, and the grids were air dried. Images were recorded on a Tecnai T12 transmission electron microscope (Thermo Fisher Scientific) operating at 120 kV with a US4000 4 K × 4 K CCD camera (Gatan) using Digital Micrograph™ software (Gatan) at a magnification of 4400x corresponding to a pixel size of 9.8 nm.

To nuclear microtubules. Based on the binding sites of some proteins already observed in vivo, one prediction might be that parasite MAPs are essential in specifying the biophysical properties of these populations[9,70]. As the preparation and analysis of cryo-FIB lamella become more routine, both higher-resolution insights into the contribution of MAPs to different microtubule architectures and larger-scale microtubule trajectories across all parasite life cycle stages can be anticipated. However, our in vitro and in vivo comparison suggests that, at least for 15-protofilament microtubules within the parasite, this can be explained solely by tubulin-intrinsic properties. In vitro microtubules with parasite-relevant architectural variants could provide an experimental framework for isolating architecture-specific microtubule-binding proteins from parasites and thereby shedding light on cytoskeleton regulatory factors. Future studies will therefore be critical in providing further insights into the diverse mechanisms by which microtubule architectures are evolutionarily tuned in a range of physiological contexts.

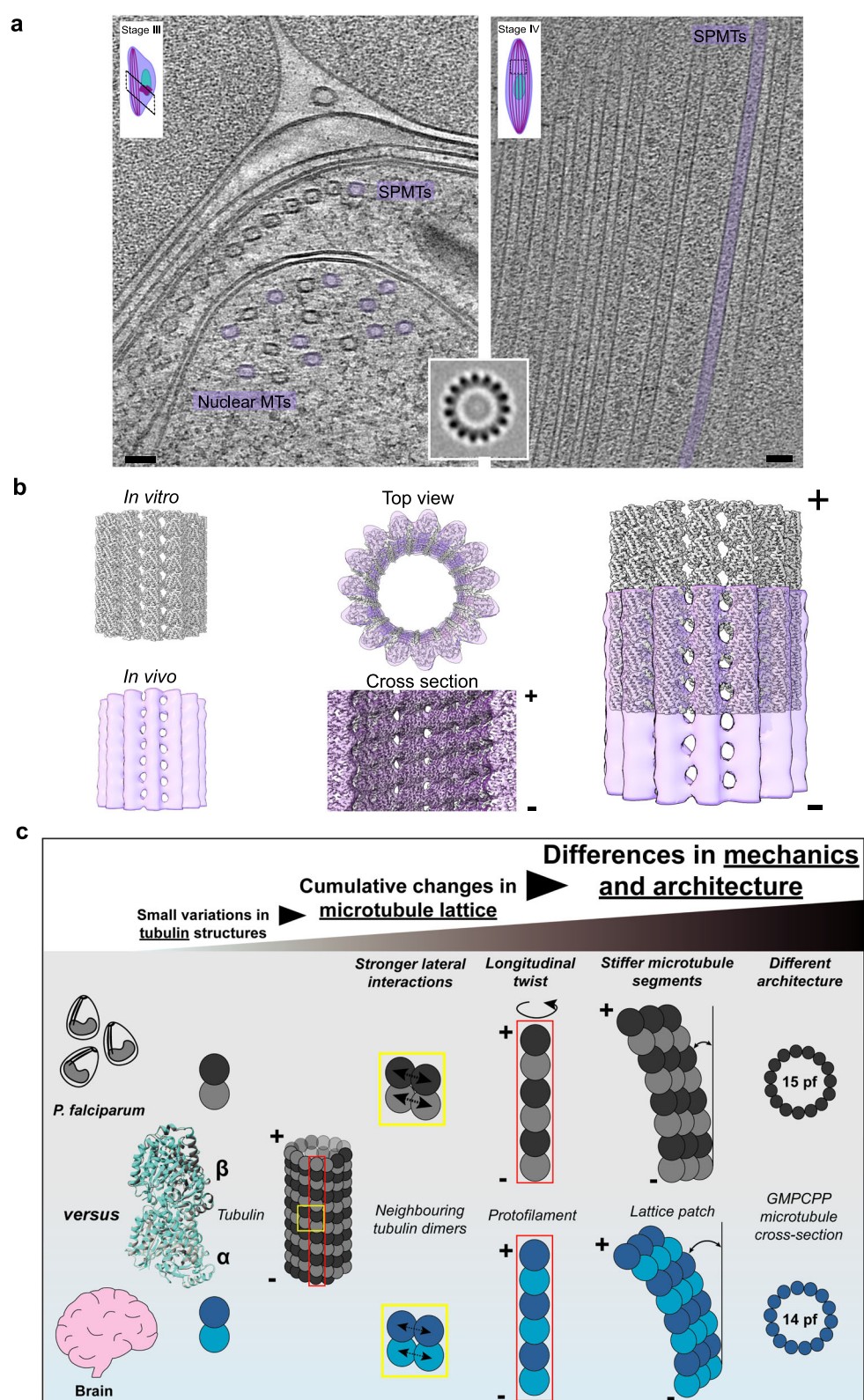

**Fig. 5 | In vitro polymerized GMPCPP microtubules are comparable to 15-protofilament microtubules in the parasite. a** Slice through tomograms of *P. falciparum* gametocytes in Stage III (left) and Stage IV (right) with 15-protofilament microtubules colored violet. SPMTs, subpellicular microtubules Scale bar = 50 nm. **Inset** shows a horizontal section through the subvolume average of a 15-protofilament microtubule **b** C1 3D reconstruction of 15-protofilament GMPCPP microtubules from single particle analysis (gray) fit into subvolume average of 15-protofilament microtubule from cryo-electron tomography of *P. falciparum* gametocytes (violet, transparent). Top, side, and cross-section views of super-imposed in vitro and in vivo microtubule structures have been shown. Plus and minus ends of the microtubules are indicated. **c** Model describing the characterized properties of *P. falciparum* tubulin compared to brain tubulin and how they can result in cumulative structural changes in the microtubule lattice that affect mechanical properties and architecture.

## Cryo-EM sample preparation

GDP-paclitaxel *P falciparum* microtubules were polymerized by incubating *P falciparum* tubulin (5 μM) with 2 mM GTP in BRB80 buffer on ice for 10 minutes followed by incubation in a water bath maintained at 37 °C. After incubation for 20 min, 5 μM paclitaxel was added to the sample, followed by further incubation in the 37 °C water bath for 1 hour before adding to EM grids for vitrification. Recombinant motor domain (MD) of *P. falciparum* kinesin-8B[12] was used to improve image alignment during subsequent 3D reconstruction of GDP-paclitaxel microtubules. Previously purified 50 μM *P. falciparum* kinesin-8B-MD was desalted, resuspended in BRB80 buffer and incubated with apyrase (10 units/mL) at room temperature for 15 min[12]. C-Flat 2/2 Holey Carbon grids (Protochips) were glow discharged in a Harrick Plasma Cleaner using air at 0.2–0.3 Torr for 40 seconds. 2 μL of GDP-paclitaxel microtubules was applied to the surface-treated grids at room temperature. After 30 seconds, the excess sample was wicked off using a filter paper and another 2.5 μL was immediately added to the grid. Excess sample was again wicked off after 30 seconds, followed by the addition of 3.5 μL of *P. falciparum* kinesin-8B-MD to the grid. After incubation for 30 seconds, 3 μL of the sample was drawn out by pipetting, and 3.5 μL of *P. falciparum* kinesin-8B-MD was added again to the grid. The grid was transferred to a humidified Vitrobot Mark IV (Thermo Fisher Scientific) chamber maintained at a temperature of 25 °C and humidity of 100 %. Excess liquid was blotted out after incubation for 30 sec, and the grid was plunge-frozen in liquid ethane.

For GMPCPP microtubules, different concentrations of *P. falciparum* tubulin (4 μM and 8 μM) were incubated with 1 mM GMPCPP in BRB80 buffer on ice for 10 minutes. The samples were then transferred to a water bath maintained at 37 °C for 2 h. EM grids were prepared by single or double application of sample. Grids with GMPCPP microtubules polymerized from 8 μM *P. falciparum* tubulin were frozen after a single application of 3 μL of sample. For grids prepared using double application of sample, 2.5 μl of GMPCPP microtubules polymerized from 4 μM *P. falciparum* tubulin were applied to the surface-treated grids at room temperature. After 30 seconds, the excess sample was wicked off using a filter paper and another 2.5 μL of the sample was immediately added to the grid again before being transferred to a pre-equilibrated Vitrobot chamber. In both cases, excess liquid was blotted out after incubation for 30 seconds, and the grids were plunge-frozen in liquid ethane.

Grids with GDP-paclitaxel *P. falciparum* microtubules without application of kinesin were also prepared by double application in the manner described above. Brain GDP-paclitaxel and GMPCPP microtubules were polymerized in a similar manner as *P. falciparum* microtubules using 10 μM bovine brain tubulin purified on a TOG column. Brain GDP paclitaxel microtubules were similarly vitrified after a single application, while brain GMPCPP microtubules were frozen following a double application.

## Cryo-EM data collection

Data were collected on a Titan Krios D3771 microscope (Thermo Fisher Scientific) operating at an accelerating voltage of 300 kV attached to a BioQuantum K3 direct electron detector (Gatan) and post-column GIF energy filter (Gatan) with a slit width of 20 eV in the .TIFF format. The samples were imaged using the automated EPU software at a nominal magnification of 81,000x resulting in a pixel size of 1.067 Å and defocus range within -0.6 to -2.4 μm. Data collection statistics for *P. falciparum* 13-protofilament GDP-paclitaxel bound to *P. falciparum* kinesin-8B-MD and 15-protofilament GMPCPP microtubules are provided in Table 1.

A total of 2968 movies were collected for *P. falciparum* GDP-paclitaxel microtubules without kinesin using the same parameters as above to further assess protofilament numbers in microtubules. 50 frames were collected for each movie with an exposure time of 3.8 sec and a total electron dose of 50 e⁻/Å². For the bovine brain microtubules dataset, 136 and 133 movies were collected for GDP-paclitaxel and GMPCPP microtubules, respectively, with an exposure time of 2.9 sec and a total electron dose of 47 e⁻/Å² fractionated over 50 frames.

## Cryo-EM image processing

The datasets of *P. falciparum* GDP-paclitaxel microtubules bound to *P. falciparum* kinesin-8B-MD and *P. falciparum* GMPCPP microtubules were processed identically. In both cases, the movies were manually screened for contaminating ice and filament quality, and poor-quality movies were discarded. RELION v3.1[72,73] and the customized Microtubule RELION-based Pipeline (MiRP)[13,14] were used to perform the following image processing steps (Supplementary Figs. 1a and 6a). Beam-induced motion of particles in movies was corrected using the inbuilt function of MotionCor2[74] in RELION. CTF estimation was performed on dose-weighted and motion-corrected summed images using CTFFIND 4.1[75] within RELION's GUI. Microtubule segments were selected manually using RELION's helical picker[76]. 4x binned particles were extracted in a box of 568 ×568 pixels with an overlapping inter-box distance of 82 Å. Supervised 3D classification using microtubule references for 11-3, 12-3, 13-3, 14-3, 14-4, 15-3, 15-4 and 16-4 protofilament architectures were used to segregate the microtubule segments (https://github.com/moores-lab/MiRPv2). 4x binned 13-protofilament GDP-paclitaxel+kinesin microtubules and 2x binned 15-protofilament GMPCPP microtubules were re-extracted for further processing. For 13-protofilament GDP-paclitaxel microtubule +kinesin segments, rotational angle and X/Y coordinate shifts were assigned based on a 15 Å low pass filtered reference of 13-protofilament microtubule decorated with kinesin motor domain (https://github.com/moores-lab/MiRPv2). Similar angles and shifts were assigned for the 15-protofilament GMPCPP microtubule segments using a 6 Å low pass filtered synthetic 15-3 protofilament microtubule generated in Chimera[77,78]. The correct assignment of the seam was verified by supervised 3D classification using all combinations of rotated and shifted references. Finally, the unbinned aligned particles were extracted for both types of datasets. For GMPCPP microtubules, aligned particles from 2 datasets were combined and used for the subsequent step of 3D reconstruction. C1 and symmetrized 3D reconstructions were obtained for the GDP-paclitaxel+kinesin and GMPCPP microtubule particles by 3D refinement runs using 15 Å and 10 Å lowpass filtered references, respectively. The quality of the electron density maps was improved by per-particle CTF refinement and Bayesian polishing in RELION. The final displayed reconstruction in Fig. 1a and Supplementary Fig. 1a corresponds to the symmetrized reconstruction of 13-protofilament GDP-paclitaxel+kinesin microtubule that was sharpened using local resolution in RELION (EMD-53571). The reconstruction shown in Fig. 4d and Supplementary Fig. 6a represents the final symmetrized reconstruction of 15-protofilament GMPCPP microtubule sharpened in RELION using its local resolution program, EMD-53572.

To assess protofilament architecture distribution, the datasets of *P. falciparum* GDP-paclitaxel microtubule without kinesin and bovine brain GDP-paclitaxel and GMPCPP microtubule samples were motion corrected and processed similarly as above. CTF estimation and particle picking were carried out on summed motion-corrected micrographs with 2x binned microtubule segments selected for extraction. The extracted particles were segregated into microtubules having different protofilament architectures using supervised 3D classification as described above. The distributions were plotted using GraphPad Prism v10.1.1 (GraphPad Software, Inc.).

## Model building, refinement and analysis

An initial model was obtained in ModelAngelo[79] using the protein sequence for *P. falciparum* tubulin (Uniprot IDs: Q6ZLZ9 for *Pf* α-tubulin, Q7KQL5 for *Pf* β-tubulin) and the symmetrized 13-protofilament GDP-paclitaxel microtubule electron density map with

a mask around the tubulin dimer in the centre of the protofilament opposite the seam. The model obtained from ModelAngelo was further checked and refined manually in Coot[80], followed by real-space refinement in PHENIX[81–83]. The model's geometry was further optimized using a combination of manual checking in Coot, interactive molecular dynamics force field-based model fitting in ISOLDE[84] and real-space refinement in PHENIX. This refined model was used as the initial model for a similar model-building and refinement pipeline for the symmetrized 15-protofilament GMPCPP microtubule electron density map.

Twelve copies of the refined tubulin dimer were fit into both GDP-paclitaxel and GMPCPP microtubule electron density maps, respectively, and refined again in PHENIX with the application of non-crystallographic symmetry restraints. These 4 tubulin dimer x 3 protofilament lattice patches were used for subsequent molecular dynamics calculations. The central tubulin dimer from each of these 4 ×3 models was deposited to the Protein Data Bank. All images were made using Chimera[77,78] or ChimeraX[85].

To compare C1 and symmetrized 3D reconstructions, copies of atomic models for the tubulin dimer were rigid-body docked into the corresponding reconstructions for a single helical turn. The superposition between the C1 and symmetrized models was carried out in the PyMOL Molecular Graphics System, Version 3.0.1, Schrödinger, LLC., and colored using the color-by-rmsd option in PyMol. The values were normalized to be on the same scale, ranging from 0 to 0.5.

## Molecular Dynamics simulations and analysis of microtubule lattice patches

In addition to the models for *P. falciparum* microtubules in the GDP-paclitaxel and GMPCPP-bound states as described in the previous section, lattice patches were obtained for porcine brain GDP-paclitaxel EMD-8322[30] and GMPCPP EMD-7973[16] microtubules. The deposited models (PDB ID: 5syf and PDB ID: 6dpu for porcine brain microtubules were expanded for twelve tubulin dimers and refined in PHENIX with non-crystallographic symmetry restraints. These atomic models were then used to construct the all-atom input structures comprising 3 protofilaments x 4 heterodimers for MD simulations. The missing loop amino acids were modeled using MODELLER[86,87] based on the corresponding UniProt sequences (Uniprot IDs: Q6ZLZ9 for *Pf* α-tubulin, Q7KQL5 for *Pf* β-tubulin, Q2XVP4 for brain α-tubulin, and P02554 for brain β-tubulin). To accommodate potential bending of the lattice patches and avoid substantial interactions with periodic images, each lattice patch was positioned in the center of a 21 nm × 20 nm × 40 nm simulation box, with the longitudinal principal axis of the lattice patch aligned along the *z*-axis of the simulation box. The system was solvated using TIP3P water and neutralized with 150 mM KCl. We parameterized the system with CHARMM36m force field[88] using CHARMM-GUI[89] and used Particle Mesh Ewald (PME) for the long-range electrostatic interactions[90]. In addition, GMPCPP was parameterized using CGenFF[91]. All the MD simulations were performed using the GROMACS 2019.6[92]. Energy minimization was performed in two stages using the gradient descent algorithm, with a step size of 0.0002 nm for the first stage and subsequent 0.002 nm for the second. The system was equilibrated at 310 K for 10 ns under constant *NVT* ensemble, followed by a 20 ns equilibration at 1 atm under constant *NPT* ensemble, with the position restraints on solute heavy atoms gradually removed from 1000 to 200 kJ·mol⁻¹·nm⁻². To mimic the bulky environment of the microtubule plus end using only 8 layers of tubulins, position restraints were applied to all heavy atoms of the three α-tubulins at the bottom-most layer during the constant *NPT* production run[32]. Canonical velocity-rescaling thermostat[93] and Parinello-Rahman barostat[94] were applied to control the temperature and pressure, respectively. Three replicas of production runs were performed for approximately 0.5 to 1 μs using the leap-frog integrator with a timestep of 2 fs. Equilibrium trajectories were collected for further analysis after RMSD

stabilization, which is conducted by the MDAnalysis package[95]. The specific procedure is detailed in the flow chart in Supplementary Fig. 4.

For each MD trajectory, twist-bending angles were calculated for each of the three protofilaments as the deviation between the vector connecting the centroids of the topmost and bottom tubulins and the straight reference axis. Because single-protofilament measurements were noisy, the three values were averaged per frame before block-averaging over time. Tangential swing was computed for the central protofilament only, using the adjacent protofilaments as reference walls to define the tangential plane. RMSD profiles depicted in Supplementary Figs. 4b and 6d mainly reflect the equilibration and overall drift of the lattice patch and were therefore used as a convergence metric. Minor deviations among replicates arise from stochastic differences in initial configurations. The differences in twist-bending and tangential swing angles provide a more direct and statistically robust measure of lattice flexibility.

## HeteroENM parameterization of lateral interactions

Each tubulin monomer was coarse-grained (CG) into a single CG bead, positioned at the center-of-mass of all the Cα atoms of the monomer. To estimate the strength of the microtubule lateral interactions, the heterogeneous elastic network model (HeteroENM) was employed for the CG representation of lattice patches[35]. The HeteroENM method can be briefly described as follows: All the pairs of CG beads are connected using harmonic springs if the equilibrium length of that pair falls within a cutoff distance of 85 Å. The spring parameters are iteratively optimized by minimizing the numerical difference between bond fluctuations calculated from the CG trajectory and those observed in the reference all-atom trajectory. By mapping fluctuations observed in the all-atom (AA) trajectory onto the CG sites, the optimized HeteroENM force field can effectively reproduce the AA-level fluctuations[36].

Absolute non-bonded interaction energies obtained from all-atom MD trajectories were not compared directly between *P. falciparum* and brain microtubule systems, because the total number of atoms and residue compositions differ between the two lattices. Such differences lead to scale-dependent total energy values that do not directly reflect inter-dimer stiffness. Instead, the HeteroENM approach provides a normalized coarse-grained representation, in which each tubulin monomer is modeled as a single bead and the spring constants are derived from correlated fluctuations observed in MD trajectories, enabling direct comparison of effective inter-dimer interaction strengths across species.

## Cryo-electron tomography data collection of microtubule ends

Cryo-electron tomography (cryo-ET) data were collected on the previously described vitrified samples of in vitro polymerized *P. falciparum* and bovine GDP-paclitaxel microtubules without kinesin using a 300 kV Titan Krios D3771 microscope (Thermo Fisher Scientific) equipped with a BioQuantum K3 direct electron detector (Gatan) and post-column GIF energy filter (Gatan). The samples were imaged using the automated EPU software Tomo5 with a slit width of 20 eV. Tilt series were collected using the dose-symmetric scheme for a range of ±60 degrees with a step size of 3 degrees. A nominal magnification of 64,000x resulting in a pixel size of 1.35 Å was used to record the datasets with a defocus range of −2.0 to −6.0 μm. For all datasets, each tilt was fractionated for six frames, and the total dose per tilt series was limited to 150 e⁻/Å². 170 and 201 tilt series were collected for *P. falciparum* and brain GDP-paclitaxel microtubule datasets, respectively.

## Cryo-electron tomogram reconstruction

Cryo-ET tilt series were processed using the tomography pipeline of RELION v5.0[72,73]. The movies were motion corrected using RELION's own implementation of MotionCorr and CTF estimation was carried out using CTFFIND 4.1 within the framework of RELION. Each of the tilt series were manually examined in Napari[96] and those containing grid

bars or large contaminating objects were removed. The remaining tilt series were aligned in a batch using the AreTomo[97] option of patch tracking provided in RELION. Reconstructed tomograms were binned to a final pixel size of 1 nm.

## Analysis of microtubule ends

Microtubule polarity in the cryo-ET data was manually ascertained by using directional protofilament skew[9,98]. Briefly, particles were picked from the microtubule end towards the microtubule lattice in IMOD[99]. Subtomogram averages were generated for the picked microtubule segment using Particle Estimation for Electron Tomography (PEET) in IMOD[100]. The microtubule end was assigned as a plus-end if the segment showed an anticlockwise skew when looking from the end towards the lattice and a minus-end if it showed a clockwise skew. This process also helped to identify the protofilament number in the averaged microtubules. To prevent bias, tomograms containing plus ends from multiple datasets of *P. falciparum* and bovine GDP-paclitaxel microtubules were merged for manual segmentation in 3dmod. Protofilaments were traced as the microtubule segments were rotated around the microtubule axis by (360°/PF number) and only longitudinal sections were used for tracing[37,38,101]. The length of the end segments as well as the curvatures were analysed from the protofilament coordinates using customized Matlab scripts (https://github.com/ngudimchuk/Process-PFs)[37,38]. GraphPad Prism v10.1.1, GraphPad Software, Inc., was used for plotting the values and statistical analyses.

## Microtubule stabilization and TIRF microscopy

Flow chambers were constructed with coverslips passivated with tri-methylchlorosilane and mounted onto passivated glass slides using thin strips of parafilm as described in ref. 11. The chamber was rinsed twice with 20 μL of 1x BRB80 and twice with a blocking buffer comprising 1% (w/v) Pluronic F-127 in 1x BRB80. The chamber was incubated in blocking buffer at room temperature for 15 minutes and washed twice with a wash buffer containing 1 mg/mL κ-casein in 1x BRB80.

Polymerization reactions were carried out at 37 °C in 1x BRB80 buffer and 2 mM Trolox, 2.5 mM Protocatechuic Acid (PCA), 25 nM Protocatechuate-3,4-dioxygenase (PCD), 0.15% methylcellulose, 1 mg/mL k-casein, and 1% β-mercaptoethanol. Purified *P. falciparum* tubulin was supplemented with 10% Atto565-labeled porcine brain tubulin, which we previously showed does not affect microtubule dynamics[11], and stabilized at 2.2 μM with 1 mM GMPCPP or at 3.3 μM with 2 mM GTP and 3 μM paclitaxel. 20 μl of the polymerization mixture was flown through the chamber, and both ends were sealed with a fast-curing silicone glue.

The slide was mounted on an inverted Nikon Eclipse Ti-E microscope with a motorized TIRF angle, a Nikon Plan Apochromat 100x/1.49NA oil immersion objective lens, and a Photometrics Prime 95B sCMOS camera, controlled by Nikon Imaging Software. Microtubules were imaged at 37 °C with a 561 nm laser with 200 ms exposure time.

## Subvolume averaging (SVA) of in vivo cryo-electron tomography data

Subvolume averaging of 15-protofilament microtubules was carried out by reprocessing FIB (focused ion beam)-milled tomograms of *P. falciparum* gametocytes from[9] EMD-15536 to 1) identify more 15-protofilament microtubules to improve the average and 2) not impose any helical symmetry. Microtubules were manually traced in IMOD[99] and initial coordinates were generated by interpolating between these points using scripts based on TEMPy[102]. Coordinates were initially over-sampled at a ~ 1 nm distance to improve signal-to-noise ratio in short microtubules, corresponding to an initial particle number 762 4 nm tubulin repeats. Each microtubule was averaged individually with the initial particle Y axis set using vectors between pairs of points along a

microtubule. To determine the number of protofilaments (and exclude all that had more or fewer than 15) as well as the microtubule polarity, initial rotations around the Y axis were randomized. Models were flipped so that all microtubules were averaged with the same polarity. PEET was used for SVA. All particles from microtubules with 15-protofilament architecture were combined (18 microtubules across 10 tomograms). SVA was performed, starting with 4x binned and moving to 2x binned, weighted-back projection data. C15 symmetry was originally imposed to increase particle number to ~1 subvolume per tubulin monomer, followed by multiple rounds of alignment and then the removal of duplicates based on low cross-correlation coefficient or overlapping positions. The final average had a particle number of 12422. It was filtered using arbitrary B-factors using Bsoft[103]. To fit the C1 map and model obtained from single particle analysis to the SVA, "Fit to Map" was used within ChimeraX[85].

## Reporting summary

Further information on research design is available in the Nature Portfolio Reporting Summary linked to this article.

## Data availability

The cryo-EM reconstructions generated in this study have been deposited in the Electron Microscopy Data Bank under accession code EMD-53571 (13 protofilament *P. falciparum* paclitaxel stabilized GDP microtubule) and EMD-53572 (15 protofilament P. falciparum GMPCPP microtubule). The corresponding structural models generated in this study have been deposited in the Worldwide Protein Data Bank under the accession codes PDB 9R4X (13 protofilament *P. falciparum* paclitaxel stabilized GDP microtubule) and PDB 9R4Y (15 protofilament P. falciparum GMPCPP microtubule). The updated subvolume average for the in vivo *P. falciparum* 15-protofilament microtubule generated in this study has been deposited in the Electron Microscopy Data Bank under accession code: EMD-56294 (subvolume average for in vivo *P. falciparum* 15-protofilament microtubule). The source data generated in this study are provided in the Source Data file. Source data are provided with this paper.

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

## Acknowledgements

We thank former and current members of the Moores and Reber laboratories for critical discussion, in particular William Hirst, who helped with initial data collection, and Tianyang Liu and Fiona Shilliday, who provided the recombinant *P. falciparum* kinesin protein. We thank the AMBIO (Charité, Berlin) for imaging support. This work was supported in part by the Max Planck Society and the Deutsche Forschungsgemeinschaft (DFG, German Research Foundation, Project-ID 528483508 – FIP 12 and RE 3925/6-1) to S.R. D.F. was supported by the Alliance Berlin Canberra co-funded by a grant from the Deutsche Forschungsgemeinschaft (DFG) for the International Research Training Group (IRTG) 2290 and the Australian National University. Cryo-EM work was supported by the Medical Research Council, U.K. (MR/R00352/1 and MR/Y000633/1 to C.A.M.) and a Wellcome Career Development award (227774/Z/23/Z to J.L.F.). Cryo-EM data were collected at the ISMB EM facility (Birkbeck College, University of London) with financial support from the Wellcome Trust (202679/Z/16/Z and 206166/Z/17/Z). The MD studies in this work were supported by the United States Department of Energy, Office of Science, Basic Energy Sciences, under Award DE-SC0023318 (to G.A.V). We thank Natasha Lukoyanova and Shu Chen for electron microscope support, David Houldershaw for computational support at Birkbeck, Ina Wagner (MPI-IB) for technical assistance, and Vladimir Volkov (Queen Mary University of London) and Nikita Gudimchuk (Lomonosov Moscow State University) for assistance with cryo-ET microtubule end analysis.

## Author contributions

M.B. performed all structural experiments, including model building. J.W. and D.B. performed and analysed molecular dynamics simulations. D.F. purified and tested tubulins. J.L.F. analyzed in vivo microtubule architecture. G.A.V. supervised molecular dynamics experiments and helped analyze the results. S.R. conceptualised the study, supervised biochemical aspects of the project, and gave critical inputs throughout. C.A.M. supervised structural data collection and analysis. M.B. and C.A.M. wrote the final manuscript with input from all authors.

## Competing interests

The authors declare no competing interests.
