## [Transparent Peer Review file · Nature Communications]

Adaptations in Plasmodium tubulin determine distinct microtubule architectures, mechanics and drug susceptibility

Corresponding Author: Professor Carolyn Moores

Version 0:

Reviewer comments:

Reviewer #1

(Remarks to the Author)

In this manuscript, Banger et al. investigate the structure of microtubules purified from Plasmodium falciparum, a malaria-causing parasite, and compare their findings with those of the more widely studied porcine or bovine microtubules. The authors present their analysis in a logical way that makes it accessible to readers, including those who are not structural biologists. Furthermore, their findings are particularly interesting for researchers studying microtubules and those interested in developing anti-parasitic therapeutics. They discovered that P. falciparum (P. fal) tubulin, despite being only ~85% identical to mammalian tubulin, forms microtubules that are structurally very similar to porcine and bovine brain tubulin. However, key residue differences lead to small shifts that appear to result in striking differences at the plus ends, the growing ends of the microtubule. They find that plus ends are less curved than the characteristic architecture of mammalian microtubules. Based on molecular dynamic simulations, the authors conclude that the structural differences they found likely result in stiffer microtubules, which would lead to a less curved plus end. Overall, this is a well-written and engaging story that will be of interest to microtubule biologists studying tubulin diversification. Specific comments below pertain mostly to clarity and further justification for claims in the manuscript.

Major comments:

1. A key finding in the paper is that P. fal tubulin exhibits less buried surface area in the alpha-alpha and beta-beta lateral interfaces. Because of this, the authors surmised this likely gives rise to different structural dynamics from mammalian tubulin and then tested this hypothesis via molecular dynamic simulations. The MD simulations demonstrated that P. fal microtubules are likely stiffer. In support of this phenomenon, mammalian tubulin also exhibits more stiffness upon reduction of buried surface area, in the case of GDP-taxol conditions versus GMPCPP-bound. This is non-intuitive: given less buried surface area, I would have guessed microtubules would be less stiff and more flexible to move. It would help the reader for the authors to explain this non-intuitive finding, both in the results and in the discussion. This may be a phenomenon that has already been characterized in the microtubule field, but it is not explicitly addressed. Furthermore, the authors claim there are no differences in longitudinal interactions, though longitudinal differences could explain the diverged architectures of the plus ends. Please show the same analysis for the longitudinal interactions as done with the lateral interactions in Figure 2.
2. The MD simulations suggest that the P. fal microtubules are stiffer, but there is no direct evidence testing this idea, beyond the observations and inferences made with the plus end structure. This idea could be tested in a couple of ways. For example, the authors could use the MD simulations to test whether mutations in the lateral interface (altering P. fal residues to the equivalent porcine or bovine tubulin residue) lead to less flexibility. Alternatively, the authors could leverage their TIRF imaging and measure the curvature of the microtubules, or persistence length (see Sweet et al. 2022 PMID 35197505; van den Heuvel and Dekker 2007 PMID 17887718). If the microtubules curve more than mammalian tubulin, then this would suggest the microtubules are indeed stiffer throughout the lattice (not just at the plus end).
3. The authors claim a couple of times that their structure provides justification for why P. fal tubulin is susceptible to oryzalin. However, this would be better substantiated with one or more mutants, perhaps tested in the MD simulations. Otherwise, I think the claims need to be attenuated (lines 413-415) since there is no actual evidence directly testing whether the shifts and residue changes are responsible for this susceptibility.
4. Throughout the manuscript, P. fal tubulin is consistently compared to "mammalian" tubulin. However, many microtubule biologists are well aware that there are multiple isoforms and residue differences, albeit small, across species. Furthermore, sometimes it is porcine tubulin that is being compared and in other cases, bovine brain tubulin. Please note the species

when referring to “mammalian” tubulin, either in the figure or the legend and text and provide justification for the isotypes used when comparing sequences.

Minor comments:

1. Fig. 1C in Extended Data: in the text, S9 is referred to (line 112) but the figure denotes S8. Also, the text refers to Fig. 1B, though they are noted in 1C. Please clarify.
2. Line 150 refers to H11', but this is not labeled in Figure 1G.
3. Line 147 refers to structural shifts in S5 and S6 that likely lead to oryzalin susceptibility, but these are not noted in Fig. 1G. Please use arrows to note what shifts are being referred to. Also, there are shifts (visible by eye) that are not noted with arrows. What is the cutoff for shifts (in Angstrom) for what is labeled with an arrow?
4. How does the overall lattice spacing in Fig. 1A compare to that of a microtubule structure from a mammalian species? It was investigated with four tubulins in a protofilament in Fig. 2, but what about the overall 13-protofilament microtubule?
5. Fig. 2A: Dimers are nicely labeled, but I do not see which C-terminal shift the authors are referring to. Please add an arrow noting this shift.
6. Please explain tangential swing.
7. A model at the end of the story would be helpful for the reader.
8. Lines 369-371: Please expand upon this a bit more for the reader to understand your point. Are the parasites known to be stiff, or less pliable, before maturation?
9. In the methods: *P. fal* tubulin was spiked with 10% Atto565 porcine tubulin to visualize by TIRF. Is there evidence that the presence of porcine tubulin does not perturb the *P. fal* microtubule structure? The GMPCPP image in Fig. 4A looks strikingly different from the GMPCPP in 4B. Can you look at pure *P. fal* tubulin by DIC instead? The authors should at least state in the legend that the *P. fal* tubulin contains 10% labeled porcine tubulin.
10. Fig. 1A: Please add a label for kinesin in blue.
11. Fig. 4G: Please add units (Ang2) to this table.
12. Figure 5: Please define “SPMTs” (stable subpellicular microtubules?) in the figure legend.

Reviewer #2

(Remarks to the Author)

Reviewer #3

(Remarks to the Author)

In this manuscript, using cryo-electron microscopy Bangera et al. determined structures of *P. falciparum* microtubules polymerized in different nucleotide states from parasite-isolated tubulins, showing that *P. falciparum* tubulins can form microtubules with non-canonical protofilament numbers that depend on the polymerization conditions. Moreover, they performed molecular dynamics simulations to analyze the flexibility of the *P. falciparum* microtubules and compared with that of mammalian microtubules. They found that the *P. falciparum* microtubules show less flexibility than the mammalian microtubules, which is due to the fact that the lateral contacts in *P. falciparum* microtubules are stronger than in the mammalian microtubules.

On the portions of the manuscript that utilize molecular dynamics simulations, the following issues should be addressed.

1. In the Methods section, it was stated that the molecular dynamics simulations were performed for both GDP-paclitaxel and GMPCPP microtubules. It was not clear whether the simulated results shown in Fig. 3 were for the GDP-paclitaxel or GMPCPP microtubules. Do the two microtubule states have the same results?
2. It is not unclear that the results of twist-bending and tangential swing angles shown in Fig. 3b and c correspond to which of the three microtubule protofilaments. It is noted that since the three protofilaments have different interaction energies with their neighboring protofilaments, the three protofilaments should have different twist-bending and tangential swing angles.
3. Using all-atom molecular dynamics simulations, the longitudinal interaction energies as well as the lateral interaction energies can also be estimated. Are these interaction energies consistent with the spring constants calculated using HeteroENM?
4. As the *P. falciparum* microtubule lattice shows lower flexible than the mammalian counterpart, it is understandable that in Extended data Fig. 4b the RMSD values for the three replicate runs of the *P. falciparum* microtubule lattice patch are evidently smaller than those for two of the three runs of the mammalian counterpart. However, it is seen that the RMSD value for one of the three runs of the mammalian counterpart is similar to those for the three replicate runs of the *P. falciparum* microtubule lattice patch. In addition, in Extended data Fig. 6d the RMSD values for the three runs of the mammalian counterpart are similar to those for the three runs of the *P. falciparum* microtubule lattice patch. The explanation of these similarities of the RMSD values should be given.
5. Fig. 3 shows twist-bending and tangential swing angles. For convenience of reading, it is better to draw the two lines that

form the twist-bending angle in the left panels of Fig. 3b and the two lines that form the tangential swing angle in the left panels of Fig. 3c.

6. On line 265, it was written that 'While the curvature of the protofilaments was similar in both species', as seen from Fig. 3i. However, the simulation data showed that the twist-bending and tangential swing angles for the *P. falciparum* microtubule are smaller than those for the mammalian counterpart. This inconsistency should be commented.

Reviewer #4

(Remarks to the Author)

The authors here examine the structure and dynamics of microtubules composed of *Plasmodium falciparum* tubulin, comparing these to the well-studied mammalian microtubules. They solve cryo-EM structures to show a few divergent residues amidst a largely conserved backbone alter overall microtubule properties. The authors identify residues positioned to change binding to some microtubule-binding ligands. Further data from structural techniques and molecular dynamics shows that these divergent residues also change polymerization properties and mechanical stiffness. Finally the authors show that a subset of microtubules in the parasite have the same protofilament architecture as those shown *in vitro*.

The cryo-EM and cryo-ET data, along with the associated biochemistry are very high quality. I am less of an expert in molecular dynamics, but I find the all-atom then coarse-grained approach to extracting dynamics at the whole microtubule level very compelling, particularly as it then provides a good explanation for changes in observed microtubule end morphology. Overall the data is well-presented and well-explained, with thorough methods and appropriate statistical tests. The conclusions on the *in vitro* level are well supported. The one conclusion I find less convincing in the manuscript's current form is the link between the *in vitro* findings and microtubules (see major comment 1).

The manuscript is very well written, particularly given the difficulty of explaining the different modes of microtubule motion in an accessible way. The manuscript also references literature appropriately.

Overall, represents an exciting, timely examination of a divergent tubulin. This will have significance in both the cytoskeleton and malaria communities. The structures provided are excellent and will allow for further investigations into drug design. This study also demonstrates how relatively small changes in sequence (85% sequence identity!) lead to intrinsic differences in the function of the protein. I feel the significance could be enhanced if the sequence and property divergence was compared to other organisms (see major comment 2; minor comment 1). Having addressed the points below this manuscript is fully deserving of publication.

Suggested improvements

Major comments:

1. I wasn't clear on the authors' conclusion that tubulin-intrinsic properties drive 15-protofilament microtubules in the parasite (438-439). I interpret this as "the observed 15-protofilament microtubules in the parasite can be explained solely by tubulin-intrinsic properties". Is that what they intended? If not please would they rephrase. I had some follow-up questions relating to this issue:

- a. Does the GDP-paclitaxel 13-pf reconstruction fit into the 13-pf microtubules observed in the gametocyte?
- b. What do the authors propose is the mechanism behind the protofilament number diversity in some life cycle stages (such as gametocytes) and not others (merozoites, where solely 13-pf microtubules are not observed)? Can the authors refer to this in the discussion? At the moment I don't think the authors refer to the fact that the "unusual range of microtubule architectures" is only observed in one life cycle stage.

2. The authors' finding that the microtubules are more stiff provide more evidence that the subpellicular microtubule network in *Plasmodium* is strong and stable. Are the divergent residues conserved amongst other Apicomplexa with subpellicular microtubules, suggesting conservation of this stiffness? How about with other organisms with a subpellicular cytoskeleton, such as kinetoplastids? Would the authors include this comparison?

Minor comments:

1. Can the authors include a discussion into other organisms with non-13 protofilament microtubules, and whether they have sequence diversity in similar spots to explain protofilament number diversity? This is reviewed well in Chaaban and Brouhard 2018 (<https://doi.org/10.1091/mbc.e16-05-0271>).

2. Can the authors speculate in the discussion what (if any) structural differences alpha-2 tubulin might provide?

3. I didn't quite understand the significance of TIRF microscopy with regard to the length of microtubules with GMPCPP, particularly given the accompanying EM data:

- a. Can the authors include how long the microtubules were polymerised for the TIRF experiments in the methods?
- b. Do the authors think "clusters of tubulin" observed are short microtubules, or aggregates? Given the fluorescently labelled tubulin is bovine (understandably), is it possible that the clusters are merely aggregated labelled bovine tubulin, or that the clusters are the result of poor polymerisation between bovine tubulin and plasmodium tubulin? Can the authors clarify these points?

Reviewer #5

(Remarks to the Author)

The authors have managed to generate novel in-vitro reconstituted cryo-EM structures of *P.falciparum* microtubules under different conditions and systematically compare them to mammalian microtubule standards. By doing this, they have developed a rationale to better understand their differential mechanics and drug susceptibility, based on their structural findings and molecular dynamic simulations. This work represents an advancement in the fields of malaria treatment, *P.falciparum* biology and microtubule molecular biology and biophysics. The work supports the conclusions drawn, although this reviewer may always take in vitro and in silico studies with a pinch of salt when attempting to draw conclusions about the in cellulo systems (which the authors do not do in the manuscript, as far as I could tell). I do not see any reason to prohibit publication of this work, although some minor points could be addressed, mostly for clarity. The methodology is sound, to the extent of my knowledge and regarding the aspects that I am supposed to review. Regarding the cryo-ET standards, a small amount of extra work may have to be carried out (please see comments below).

Fig 1a caption, please specify again the organism of the tubulin used. Also, in Fig 1a, there seems to be a something not too clear as the caption reads “a...GTP-paclitaxel (left)” while the label on the image on the left in a says “GDP-paclitaxel”. In Fig 1c the same GDP to GTP change shows up. It can be understood that microtubules will hydrolyze the GTP into GDP, and what is written is correct. Please consider clarifying this point to make the labels on images and the description on the captions more similar.

In Extended Data Figure 1 a and EDFig6a, it also would be very informative to have the particle number for each class (or percentage) in the 3D classifications.

In Fig5 and EDFig5, please add the number of subtomograms used for subtomogram averaging (this cannot be deduced from microtubule number and sampling distance as we do not know the length of each individual microtubule). It would be clear how many particles were picked and used for both in vitro and in vivo averages.

In the Methods section, when the EM movies are recorded, it would be nice to specify their format. I am assuming it is .mrc

Each structure, independently of its resolution and software used should be deposited (I only see two depositions in the review data, while there are other novel structures coming from this work). There is no report on resolution, angular distribution and other metrics for the in vivo average present in Fig5 and EDFig7. This information seems to be missing also for the in vitro STA of microtubules present in EDFig5. Any EM map, independently of the resolution, should be processed as to be deposited in the EMDB, according to the latest standards (I believe that latest PEET versions do support half-map creation, but they also can be obtained by properly splitting model and motive list files into two random datasets and applying the same .epe particle alignment run to both). RELION can also be used for STA, giving normally better results than PEET and also all sorts of metrics to assess the alignment quality

Please indicate the number of micrographs acquired for each dataset in Table 1 (it gives an idea of the microtubule density, particle quality, and other metrics)

In Fig 2, the following can be read: “Further, analysis of the residue composition of the H3 helix 206 revealed an increase in the ratio of non-polar to polar residues at the β - β interface in *P. 207 falciparum* microtubules (Figs. 2c and d; Extended Data Figs. 2a and b).” According to the number of polar and non-polar residues in the figure, the non-polar to polar ratio should decrease, not increase. It would increase if it would be the polar to non-polar ratio. Please only refer to Fig 2 d for β - β interaction.

In Fig 3 g, please include a crosssectional view of the microtubules in the tomogram to illustrate how easy was it to trace individual protofilaments. ISONET-based filtering may help mitigating missing-wedge artifacts for visualization.

I would like to suggest the authors, with this or further work, to pursue the experimental elucidation by cryo-EM of the binding site of oryzalin. Also, I believe it is possible to easily reach subnanometric resolution of in vivo microtubules when having access to enough high-quality data from cryo-FIB SEM samples. Although the MD based rigidity assessment is credible, there could be another ways to measure the rigidity of these microtubules: Bending rigidity estimation in partially clamped microtubule gliding assays, optical tweezer approaches or subjection of microtubules to fluid flows in controlled microfluidic chambers.

Version 1:

Reviewer comments:

Reviewer #1

(Remarks to the Author)

The authors strengthened the manuscript by adding the model and measurements of the longitudinal contact surface area, thereby rounding out their argument for the critical change in lateral contacts. A couple of points follow:

Author Response 3 and 4: While direct testing of stiffness would have significantly strengthened your arguments, I understand it might take months to optimize a gliding assay. Thank you for toning down the language in the manuscript. However, the abstract should also explicitly state that the structure and MD simulations suggest increased stiffness and how

P. fal microtubules differ in drug susceptibility. The current language implies that the structure proves these features (lines 37-38).

Author Response 6: "However, we have previously shown that at the low concentration used, the presence and source of labelled tubulin does not affect microtubule dynamics (Hirst et al., 2020)." Please mention this statement and reference in the figure legend or methods.

Reviewer #3

(Remarks to the Author)

The authors have addressed my comments on the manuscript.

Reviewer #4

(Remarks to the Author)

The authors have addressed all my points well. As such I would now fully support this for publication. This is an exciting article that will have significance in both the cytoskeleton and malaria communities.

To address the one remaining point from the authors:

a. Does the GDP-paclitaxel 13-pf reconstruction fit into the 13-pf microtubules observed in the gametocyte?

Response 25: Yes, it does fit but the resolution of the subvolume average is low and, because there are intrinsically fewer structural features in 13-protofilament microtubules (the protofilaments are straight so there is no moiré pattern), the comparison is less informative. For this reason, we would not propose to include the 13-protofilament microtubule data in the revised manuscript but would be happy to be guided by the reviewer/editor on this point.

The rebuttal figure looks convincing at the resolution, though I agree that this need not to be included.

Reviewer #5

(Remarks to the Author)

Manuscript#: NCOMMS-25-60374A Reviewer: #5

Corresponding Author: Carolyn A. Moores Due Date: 27th December 25

Title: Adaptations in Plasmodium tubulin determine unique microtubule architectures, mechanics and drug susceptibility

Answer to Rebuttal (24/12/25)

General Comments:

It may be beneficial for the manuscript to come up with abbreviations for combinations of words such as 'microtubules', 'brain', 'porcine', 'bovine' and so on.

The authors have answered all my comments, however there are some points that I find crucial to be revisited. If responses 34 to 35 are not revisited. I would not be able to endorse the publication of this work.

I strongly encourage the authors to pursue experimental validation of their MD predictions by other techniques and to consider other technologies such as crosslinking mass spectrometry to investigate the binding site of oryzalin to tubulin

By-response answers:

33: Yes, it should always be clear in the text to which organism the structures correspond together with the experimental conditions used (in silico, in vitro, in cellulo, etc). When I cite the following piece of your manuscript: "a.[...]GTP-paclitaxel (left)", this corresponds to Figure 4 caption, not Figure 1, mea culpa. Still, while the label on the image on the left in a says "GDP-paclitaxel". The same apparent mistake can be found in Figure 4 c. I am still not sure if this change in nomenclature is due to the GTP hydrolysis process or just a mislabeling. Please check

34: Thanks for adding the percentages. I still have a serious concern about the 0% classes in Extended Data Fig 6 a. It is very confusing, if there are 0% of the particles in a class, how where the class averages generated for that class? When comparing them to Extended Data Figure 1a the images do not match.

35: The changes in lines 832-833 are insufficient to allow the reader to easily understand the total microtubule length used for STA. The initial sampling distance along the microtubules remains undisclosed as it is veiled in the following text: "... initial coordinates were generated by interpolating between these points using scripts based on TEMPy102". At least it should be mentioned how many tubulin-repeats (~4nm) were picked. Furthermore, after duplicate removal and similar processing, the final number of particles is not disclosed in the corresponding Materials and Methods section. Please include this information. In Extended Data Figure 5b, the "brain" label could read 'bovine' or something like that to ease the interspecies comparison for the reader. Extended Data Figure 5b has now particle number estimations. However, the image under the label "Average 3D volume" comes from a specific average, with specific particle numbers, please disclose that number.

36: Thank you for addressing the comment.

37: Thank you for addressing the comment.

38: Thank you for addressing the comment.

39: Thank you for addressing the comment.

40: I do appreciate that the authors included the microtubule crosssections in Extended Data Figure 5b. However, the crosssections do not correspond to the protofilaments in the "curved ends". Could the authors show a crosssection of these regions to illustrate how easy it was to trace the protofilaments? Was a combination of crosssections and longitudinal sections used to trace the protofilaments?

41: I would like to add that cross-linking mass spectrometry can shed light onto the tubulin binding site of oryzalin also in vivo

Reviewer #1

In this manuscript, Bangera et al. investigate the structure of microtubules purified from Plasmodium falciparum, a malaria-causing parasite, and compare their findings with those of the more widely studied porcine or bovine microtubules. The authors present their analysis in a logical way that makes it accessible to readers, including those who are not structural biologists. Furthermore, their findings are particularly interesting for researchers studying microtubules and those interested in developing anti-parasitic therapeutics. They discovered that P. falciparum (P. fal) tubulin, despite being only ~85% identical to mammalian tubulin, forms microtubules that are structurally very similar to porcine and bovine brain tubulin. However, key residue differences lead to small shifts that appear to result in striking differences at the plus ends, the growing ends of the microtubule. They find that plus ends are less curved than the characteristic architecture of mammalian microtubules. Based on molecular dynamic simulations, the authors conclude that the structural differences they found likely result in stiffer microtubules, which would lead to a less curved plus end. Overall, this is a well-written and engaging story that will be of interest to microtubule biologists studying tubulin diversification. Specific comments below pertain mostly to clarity and further justification for claims in the manuscript.

Major comments:

1. A key finding in the paper is that *P. fal* tubulin exhibits less buried surface area in the alpha-alpha and beta-beta lateral interfaces. Because of this, the authors surmised this likely gives rise to different structural dynamics from mammalian tubulin and then tested this hypothesis via molecular dynamic simulations. The MD simulations demonstrated that *P. fal* microtubules are likely stiffer. In support of this phenomenon, mammalian tubulin also exhibits more stiffness upon reduction of buried surface area, in the case of GDP-taxol conditions versus GMPCPP-bound. This is non-intuitive: given less buried surface area, I would have guessed microtubules would be less stiff and more flexible to move. It would help the reader for the authors to explain this non-intuitive finding, both in the results and in the discussion. This may be a phenomenon that has already been characterized in the microtubule field, but it is not explicitly addressed.

Response 1: Thank you for this comment. Indeed, published data on lateral contacts is ambiguous. Weakening of lateral contacts has been linked to the accumulation of lattice strain following GTP hydrolysis (Manka and Moores, 2018) but has also been proposed as a mechanism for increasing microtubule stability at sites of damage (Eshun-Wilson et al., 2019). Our own data on *Xenopus* tubulins recently suggested that *X. laevis* microtubules have the weakest lateral interactions, whereas our dynamic data suggest that they have the most stable lattice, consistent with the idea that weakened lateral contacts can contribute to overall stability (Troman et al., 2025). Additionally, simulation data have suggested a link between loop ordering within the free tubulin dimer and the activation energy required for lattice incorporation (Chaaban et al., 2018).

While we indeed find the buried surface areas in the alpha-alpha and beta-beta lateral interfaces to be smaller in *P. falciparum* microtubules, we find the lateral interactions to be stronger (lines 251-255). In other words, the buried surface area does not correlate with interaction strength in *P. falciparum* microtubules. One explanation that we explore is the composition of residues at, and adjacent to, the interface (lines 210-

219). While there is a set of residue variations in this lattice interface, it is the MD calculations that allow us to investigate the overall mechanical properties of a patch of lattice and uncover the comparative stiffness of the polymers. As suggested by the reviewer, we have now included a line in the Discussion about this (lines 443-447).

Furthermore, the authors claim there are no differences in longitudinal interactions, though longitudinal differences could explain the diverged architectures of the plus ends. Please show the same analysis for the longitudinal interactions as done with the lateral interactions in Figure 2.

Response 2: The buried surface areas for longitudinal interactions in 13-protofilament mammalian (which we now refer to as brain as per Response 5) and *P. falciparum* lattices are similar, shown in the Table below. We have added a line about this (line 183) and also included the Table in Supplementary Fig. 3a. The similarity in the strength of longitudinal interactions as estimated by MD simulations has been described in lines 251-255.

Longitudinal buried surface area	P. falciparum	Brain
Intradimer (\AA^2)	1841.8	1841.8
Interdimer (\AA^2)	1672.2	1669.0

*2. The MD simulations suggest that the *P. fal* microtubules are stiffer, but there is no direct evidence testing this idea, beyond the observations and inferences made with the plus end structure. This idea could be tested in a couple of ways. For example, the authors could use the MD simulations to test whether mutations in the lateral interface (altering *P. fal* residues to the equivalent porcine or bovine tubulin residue) lead to less flexibility. Alternatively, the authors could leverage their TIRF imaging and measure the curvature of the microtubules, or persistence length (see Sweet et al. 2022 PMID 35197505; van den Heuvel and Dekker 2007 PMID 17887718). If the microtubules curve more than mammalian tubulin, then this would suggest the microtubules are indeed stiffer throughout the lattice (not just at the plus end).*

Response 3: Again, we thank the reviewer for this important comment. While the simulation includes both lateral and longitudinal contacts of the microtubule lattice, tubulin's complex allostery means that it is not feasible to exhaustively test the origin of stiffness via mutagenesis of combinations of differences between *P. falciparum* and brain sequences in MD calculations. Furthermore, it is difficult to predict how these measurements will translate into a 10 μm microtubule. Therefore, we agree with the reviewer that it is important to measure the bending stiffness. As suggested, we went back to our TIRF assays, however, the *P. falciparum* microtubules observed in our experiment do not reach sufficient lengths to exhibit noticeable curvature. Unfortunately, we are limited by the final concentration of *P. falciparum* tubulin, which we cannot purify and concentrate to $> 20 \mu\text{M}$ (as a comparison: brain tubulin can be concentrated to $> 100 \mu\text{M}$). Alternatively, we could derive microtubule persistence length via thermal fluctuations (as in Pampaloni et al., 2006), but again microtubules need to be rather long for these assays. Therefore, we need more sophisticated assays such as kinesin gliding assays, which we have not yet set up in our lab although these are important experiments to perform in the future. To take this comment into consideration, we have taken care to avoid making overly broad claims and have

expanded the Discussion, see lines 441-447 and 453-456 (see also a more detailed discussion on mechanical properties in lines 273-288).

3. The authors claim a couple of times that their structure provides justification for why *P. fal* tubulin is susceptible to oryzalin. However, this would be better substantiated with one or more mutants, perhaps tested in the MD simulations. Otherwise, I think the claims need to be attenuated (lines 413-415) since there is no actual evidence directly testing whether the shifts and residue changes are responsible for this susceptibility.

Response 4: We have experimental evidence that oryzalin has a stronger effect on *Plasmodium* microtubules than on human (mammalian) microtubules as published in Hirst et al., 2021 (Rebuttal Figure 1 below). However, we take the reviewer's point that the mechanism for this selectivity is inferred from our structural data. As noted above, tubulin's complex allostery means it is not straightforward to test drug binding via mutagenesis, so according to the reviewer's suggestion, we toned down our claims. Please see lines 429-432, and further related discussion in Response 41 below.

Rebuttal Figure 1. Extracted from Hirst et al., 2021 demonstrating the selective inhibition by oryzalin of *P. falciparum* tubulin growth in vitro and parasite proliferation.

4. Throughout the manuscript, *P. fal* tubulin is consistently compared to “mammalian” tubulin. However, many microtubule biologists are well aware that there are multiple isoforms and residue differences, albeit small, across species. Furthermore, sometimes it is porcine tubulin that is being compared and in other cases, bovine brain tubulin. Please note the species when referring to “mammalian” tubulin, either in the figure or the legend and text and provide justification for the isotypes used when comparing sequences.

Response 5: We thank Reviewer 1 for raising this very important point. For the sake of clarity in the main text and given that the focus of the work is our discoveries relating to *P. falciparum* tubulin, we now consistently use the term ‘brain’ tubulin for tubulin obtained from mammalian brains. We have explicitly stated this in lines 117-122 but provide details of species and isotypes in the figure legends. In structural comparisons with published “brain” tubulin data, the isotype sequence is defined by the authors in the deposited coordinates, albeit with some ambiguity because regions of tubulin that enable differentiation between isoforms correspond to regions of structural flexibility that cannot be verified experimentally. Importantly though, for the purposes of our discoveries, our *P. falciparum* tubulin is isotype-pure as it only contains α 1- and β -tubulin (see below, as published in Hirst et al. 2022).

Gene	Uniprot ID	Predicted mass	Measured mass	Mass deviation
α 1-tubulin	Q6ZLZ9	50,296.97 Da	50,295.6 Da	27 ppm
α 2-tubulin	Q8IFP3	49,691.37 Da	-	-
β -tubulin	Q7KQL5	49,750.95 Da	49,749.7 Da	25 ppm

Rebuttal Figure 2. Extracted from Hirst et al., 2022 demonstrating the isotype-purity of the P. falciparum tubulin used in our in vitro studies.

Minor comments:

1. Fig. 1C in Extended Data: in the text, S9 is referred to (line 112) but the figure denotes S8. Also, the text refers to Fig. 1B, though they are noted in 1C. Please clarify.

Response 6: Lines 111-112 refer to the differences in S9-S10 loop between α and β tubulin which is commonly used as an indicator to show the distinction between α and β tubulin in a 3D reconstruction of a microtubule. This is now labelled in the Extended Data Figs. 1b and 6b. A representative section of the density map around β -strands S8 and S10 is depicted in Extended Fig. 1c to show the quality of the map.

2. Line 150 refers to H11', but this is not labeled in Figure 1G.

Response 7: H11' helix has been labeled in Fig. 1g

3. Line 147 refers to structural shifts in S5 and S6 that likely lead to oryzalin susceptibility, but these are not noted in Fig. 1G. Please use arrows to note what shifts are being referred to. Also, there are shifts (visible by eye) that are not noted with arrows. What is the cutoff for shifts (in Angstrom) for what is labeled with an arrow?

Response 8: We have added more labels in Fig. 1f to depict the interaction of the Gly-rich loop with helix H5 and strands S5 and S6. In Fig. 1g we have added arrowheads colour-coded to depict the range of RMSD values in Å (Blue: 0.0 to 1.3; Red: 1.3 to 2.5). This has been explained in the Figure legend as well.

4. How does the overall lattice spacing in Fig. 1A compare to that of a microtubule structure from a mammalian species? It was investigated with four tubulins in a protofilament in Fig. 2, but what about the overall 13-protofilament microtubule?

Response 9: The spacing in the model reflects the lattice spacing in the 3D reconstruction of microtubules. We have now added labels in Extended Data Figs 1a and 6a to indicate how the model was obtained. We had already compared the longitudinal distances between monomers which reflect the overall lattice spacing of the data averaged to produce our reconstruction in lines 181-183 and lateral interactions in lines 195-215 and Figs. 2c and d. We hope that this was what the reviewer was referring to.

5. Fig. 2A: Dimers are nicely labeled, but I do not see which C-terminal shift the authors are referring to. Please add an arrow noting this shift.

Response 10: Red arrowheads have been added to indicate shifts in the C-terminal domain

6. Please explain tangential swing.

Response 11: We now more clearly define “twist-bending” as the angle between the reference axis and the measured bent protofilament axis and “tangential swing” as the angle between the lateral reference plane and the measured central protofilament axis. We have more clearly annotated the figure panels illustrating both ‘twist bending’ as well as ‘tangential swing’ and now refer to them in the text when these terms are introduced (line 240 and Figures 3b and c). In the revised Fig. 3b, solid and dashed lines now show the reference straight axis and the bent protofilament axis defining the twist-bending angle. In the revised Fig. 3c, arrows and lines indicate the lateral reference plane and the central protofilament axis that form the tangential swing angle.

7. A model at the end of the story would be helpful for the reader.

Response 12: We have now included a model outlining the key findings from the study. Please see Fig. 5c, which we refer to on line 403.

8. lines 369-371: Please expand upon this a bit more for the reader to understand your point. Are the parasites known to be stiff, or less pliable, before maturation?

Response 13: There is strong evidence that gametocytes are stiff until Stage IV, after which their microtubules depolymerize. However, the extent to which microtubules and microtubule architecture contribute to the overall bulk stiffness of the parasite remains to be quantified. We have added a line (lines 374-376) and additional references to support this important point.

9. In the methods: P. fal tubulin was spiked with 10% Atto565 porcine tubulin to visualize by TIRF. Is there evidence that the presence of porcine tubulin does not perturb the P. fal microtubule structure? The GMPCPP image in Fig. 4A looks strikingly different from the GMPCPP in 4B. Can you look at pure P. fal tubulin by DIC instead? The authors should at least state in the legend that the P. fal tubulin contains 10% labeled porcine tubulin.

Response 14: First of all, it is important to mention that all the EM data, including those used to calculate the 3D structures involved **no labelled tubulin**. Only the TIRF data presented used 10% labelled porcine tubulin. However, we have previously shown that at the low concentration used, the presence and source of labelled tubulin does not affect microtubule dynamics (Hirst et al., 2020). Importantly, we now routinely use interference reflection microscopy (IRM), which solely relies on the interference of light reflected from microtubules and therefore does not require fluorescently labelled tubulin. Below, we show two representative figures of *P. falciparum* microtubules stabilised with either paclitaxel or GMPCPP imaged by IRM. Without labelled tubulin, we see the same effect in that paclitaxel stabilizes microtubules while GMPCPP leads to clusters of small microtubule seeds. Microtubule clusters unfortunately generate more complex interference patterns and shadowing effects, reducing contrast and making individual features harder to resolve when compared to single filaments. The same issue would also impact DIC imaging. Generally, it is known that the effect of GMPCPP on microtubule length is concentration dependent (Hyman et al, 1992) but the effect will also depend on tubulin critical concentration and growth. In other words, the detailed effects of GMPCPP will be species-specific. We have updated the text (lines 302-303) and the relevant figure legend to make this clearer.

3.3 μM *P. falciparum* tubulin stabilized with 3 μM paclitaxel

2.2 μM *P. falciparum* tubulin 1 mM GMPCPP

Rebuttal Figure 3. Interference reflection microscopy (IRM) images of P. falciparum tubulin grown in vitro and stabilized by paclitaxel (left) and GMPCPP (right).

37°C in 1x BRB80 buffer, 2.5 mM Protocatechuic Acid (PCA), 25 nM Protocatechuate-3,4-dioxygenase (PCD), 0.15% methylcellulose, 1 mg/mL k-casein, and 1% β -mercaptoethanol, image taken after 10 min after reaction assembly. Scale bar 10 μm .

10. Fig. 1A: Please add a label for kinesin in blue.

Response 15: Kinesin has been labeled

11. Fig. 4G: Please add units (Ang2) to this table.

Response 16: Buried surface area (\AA^2) has been added to the table

12. Figure 5: Please define "SPMTs" (stable subpellicular microtubules?) in the figure legend.

Response 17: The abbreviation has been added in the Fig. 5 legend

Reviewer #2

We thank this reviewer for contributing to the peer review of our manuscript.

Reviewer #3

*In this manuscript, using cryo-electron microscopy Banger et al. determined structures of *P. falciparum* microtubules polymerized in different nucleotide states from parasite-isolated tubulins, showing that *P. falciparum* tubulins can form microtubules with non-canonical protofilament numbers that depend on the polymerization conditions. Moreover, they performed molecular dynamics simulations to analyze the flexibility of the *P. falciparum* microtubules and compared with that of mammalian microtubules. They found that the *P. falciparum* microtubules show less flexibility than the mammalian microtubules, which is due to the fact that the lateral contacts in *P. falciparum* microtubules are stronger than in the mammalian microtubules. On the portions of the manuscript that utilize molecular dynamics simulations, the following issues should be addressed.*

1. In the Methods section, it was stated that the molecular dynamics simulations were performed for both GDP-paclitaxel and GMPCPP microtubules. It was not clear whether the simulated results shown in Fig. 3 were for the GDP-paclitaxel or GMPCPP microtubules. Do the two microtubule states have the same results?

Response 18: We thank the reviewer for highlighting this point. The simulation results in Fig. 3 correspond to GDP-paclitaxel microtubules while the results from MD simulations on GMPCPP microtubules are depicted in Fig. 4. We have now clarified this in line 226 and Fig. 3a legend. Interestingly, the behaviour of *P. falciparum* tubulin is qualitatively similar in each nucleotide state, although the relative properties of what we now refer to as ‘brain’ tubulin are different. These points are described in lines 324-329 and depicted in Fig. 4e.

2. It is not unclear that the results of twist-bending and tangential swing angles shown in Fig. 3b and c correspond to which of the three microtubule protofilaments. It is noted that since the three protofilaments have different interaction energies with their neighboring protofilaments, the three protofilaments should have different twist-bending and tangential swing angles.

Response 19: Thank you for pointing this out. For the twist-bending angle, we analyzed each of the three protofilaments in the simulated lattice patch separately. At each MD step, the bending of a protofilament was defined as the deviation of the vector connecting the centroids of the topmost and the bottom (position-restrained) tubulins from the initial straight axis. This yields three twist-bending values per MD timestep. Because the single-protofilament measurements are relatively noisy due to local thermal fluctuations and limited sampling, we averaged the three protofilament values within each frame and then averaged over time and three replicate runs to obtain the mean and standard deviation reported in Fig. 3b. For the tangential swing angle, we analyzed only the central protofilament of the patch, as this metric requires the neighboring protofilaments on each side to define the tangential reference plane (or the reference wall). Please see Methods section, lines 729-739, where we further clarify these points including how the angular quantities were defined and why the averaged values are presented for clarity and statistical robustness.

3. Using all-atom molecular dynamics simulations, the longitudinal interaction energies

as well as the lateral interaction energies can also be estimated. Are these interaction energies consistent with the spring constants calculated using HeteroENM?

Response 20: We thank the reviewer for this constructive suggestion. We agree that comparing inter-dimer interaction energies derived from all-atom molecular dynamics simulations with the effective spring constants derived from the HeteroENM analysis would be conceptually valuable. However, such comparisons are not quantitatively meaningful across *Plasmodium* and mammalian brain systems in the all-atom framework. This is because the two lattices contain different numbers of atoms and distinct residue compositions, resulting in absolute interaction energies that scale with system size rather than directly reflecting mechanical stiffness. In contrast, the coarse-grained HeteroENM approach treats each tubulin monomer as a single bead and parameterizes the effective spring constants as an intensive property from the relative fluctuations observed in the MD trajectories. This procedure normalizes system size and enables a consistent comparison of effective interaction strengths between the two species.

To address this point, we have clarified the rationale for using the coarse-grained model. Please see Methods section, lines 754-761 where we aim to explain why the HeteroENM analysis provides a consistent and physically interpretable metric for comparing inter-dimer interactions between species.

*4. As the *P. falciparum* microtubule lattice shows lower flexibility than the mammalian counterpart, it is understandable that in Extended data Fig. 4b the RMSD values for the three replicate runs of the *P. falciparum* microtubule lattice patch are evidently smaller than those for two of the three runs of the mammalian counterpart. However, it is seen that the RMSD value for one of the three runs of the mammalian counterpart is similar to those for the three replicate runs of the *P. falciparum* microtubule lattice patch. In addition, in Extended data Fig. 6d the RMSD values for the three runs of the mammalian counterpart are similar to those for the three runs of the *P. falciparum* microtubule lattice patch. The explanation of these similarities of the RMSD values should be given.*

Response 21: We thank the reviewer for this detailed observation. As shown in Fig. 3b and c, the twist-bending and tangential swing angles of the GDP-paclitaxel lattices differ between *P. falciparum* and brain microtubules, consistent with the overall trend in the RMSD plots. Regarding the reviewer's observation that one mammalian trajectory exhibits an RMSD similar to those of the *P. falciparum* runs, this arises from stochastic differences among individual simulations. To mitigate such randomness, we performed three independent replicates for each system and reported averaged quantities with their statistical variation.

In contrast, for the GMPCPP lattices (Fig. 4e), the difference is smaller, which is also consistent with the RMSD patterns in Supplementary Fig. 6d. We note, however, that RMSD does not exclusively reflect protofilament bending. It also includes contributions from lattice expansion and global drift that occur during equilibration. Therefore, RMSD was used only as a measure of simulation convergence, while the collective-variable analysis (twist-bending and tangential swing) provides a more direct and quantitative assessment of lattice flexibility. We have now clarified this in the Methods section (lines 734-739).

5. Fig. 3 shows twist-bending and tangential swing angles. For convenience of reading, it is better to draw the two lines that form the twist-bending angle in the left panels of Fig. 3b and the two lines that form the tangential swing angle in the left panels of Fig. 3c.

Response 22: We thank the reviewer for this helpful suggestion. We have revised Fig. 3b and c to include the lines forming the two angles that explicitly show how each angle is defined. In the revised Fig. 3b, solid and dashed lines now show the reference straight axis and the bent protofilament axis defining the twist-bending angle. In the revised Fig. 3c, arrows and lines indicate the lateral reference plane and the central protofilament axis that form the tangential swing angle.

6. On line 265, it was written that ‘While the curvature of the protofilaments was similar in both species’, as seen from Fig. 3i. However, the simulation data showed that the twist-bending and tangential swing angles for the *P. falciparum* microtubule are smaller than those for the mammalian counterpart. This inconsistency should be commented.

Response 23: We agree that this might initially seem paradoxical but we tried to reconcile the cryo-ET and MD data in the context of other recent work in the field in lines 273-278. In essence, because the MD analyses were undertaken using a lattice patch while the cryo-ET revealed the behaviour of individual protofilaments, we aimed to reconcile these data, also inspired by recent work in the field (Kalutskii M. et al., 2025). We apologise that this wasn’t sufficiently clear and we have reworked this paragraph as well as explicitly alluded to it in the Discussion (lines 453-456) to try and make it more understandable.

Reviewer #4

The authors here examine the structure and dynamics of microtubules composed of Plasmodium falciparum tubulin, comparing these to the well-studied mammalian microtubules. They solve cryo-EM structures to show a few divergent residues amidst a largely conserved backbone alter overall microtubule properties. The authors identify residues positioned to change binding to some microtubule-binding ligands. Further data from structural techniques and molecular dynamics shows that these divergent residues also change polymerization properties and mechanical stiffness. Finally the authors show that a subset of microtubules in the parasite have the same protofilament architecture as those shown in vitro.

The cryo-EM and cryo-ET data, along with the associated biochemistry are very high quality. I am less of an expert in molecular dynamics, but I find the all-atom then coarse-grained approach to extracting dynamics at the whole microtubule level very compelling, particularly as it then provides a good explanation for changes in observed microtubule end morphology. Overall the data is well-presented and well-explained, with thorough methods and appropriate statistical tests. The conclusions on the in vitro level are well supported. The one conclusion I find less convincing in the manuscript’s current form is the link between the in vitro findings and microtubules (see major comment 1).

The manuscript is very well written, particularly given the difficulty of explaining the

different modes of microtubule motion in an accessible way. The manuscript also references literature appropriately.

Overall, represents an exciting, timely examination of a divergent tubulin. This will have significance in both the cytoskeleton and malaria communities. The structures provided are excellent and will allow for further investigations into drug design. This study also demonstrates how relatively small changes in sequence (85% sequence identity!) lead to intrinsic differences in the function of the protein. I feel the significance could be enhanced if the sequence and property divergence was compared to other organisms (see major comment 2; minor comment 1). Having addressed the points below this manuscript is fully deserving of publication.

Suggested improvements

Major comments:

1. I wasn't clear on the authors' conclusion that tubulin-intrinsic properties drive 15-protofilament microtubules in the parasite (438-439). I interpret this as "the observed 15-protofilament microtubules in the parasite can be explained solely by tubulin-intrinsic properties". Is that what they intended? If not please would they rephrase.

Response 24: Indeed this is what we meant and have rephrased the text accordingly (line 466-468).

I had some follow-up questions relating to this issue:

a. Does the GDP-paclitaxel 13-pf reconstruction fit into the 13-pf microtubules observed in the gametocyte?

Response 25: Yes, it does fit but the resolution of the subvolume average is low and, because there are intrinsically fewer structural features in 13-protofilament microtubules (the protofilaments are straight so there is no moiré pattern), the comparison is less informative. For this reason, we would not propose to include the 13-protofilament microtubule data in the revised manuscript but would be happy to be guided by the reviewer/editor on this point.

*Rebuttal Figure 4. Symmetrized 3D reconstruction of 13-protofilament GDP-paclitaxel microtubule from single particle analysis (grey) fit into subvolume average of 13-protofilament microtubule from cryo-electron tomography of *P. falciparum* gametocytes (dark blue, transparent).*

b. What do the authors propose is the mechanism behind the protofilament number diversity in some life cycle stages (such as gametocytes) and not others (merozoites, where solely 13-pf microtubules are not observed)? Can the authors refer to this in the discussion? At the moment I don't think the authors refer to the fact that the "unusual range of microtubule architectures" is only observed in one life cycle stage.

Response 26: The mechanism for determining the architecture of microtubules is complex and not clearly delineated yet. From this work, we identify that *P. falciparum* tubulin has a propensity to form microtubules with different architectures. However, within the parasite, there might exist a second layer of regulation comprising α -tubulin isoforms, post-translational modifications, MAPs and other nucleating factors expressed during different stages of the parasite life cycle. We have included a line about this in Discussion (lines 396-398).

2. The authors' finding that the microtubules are more stiff provide more evidence that the subpellicular microtubule network in Plasmodium is strong and stable. Are the divergent residues conserved amongst other Apicomplexa with subpellicular microtubules, suggesting conservation of this stiffness? How about with other

organisms with a subpellicular cytoskeleton, such as kinetoplastids? Would the authors include this comparison?

Response 27: We thank the reviewer for this suggestion. To consider it, we chose a subset of organisms with subpellicular microtubules and compared them to porcine brain tubulin. Aligning α -tubulin from pig (Q2XVP4) with *P. falciparum* (Q8IFP3, Q6ZLZ9), *Toxoplasma gondii* (P10873, A0A7J6K2V4, A0A7J6K462), and *Trypanosoma brucei brucei* (Q4GYY5_TRYB2), or β -tubulin from pig (Q2XVP4) with *P. falciparum* (Q7KQL5), *Toxoplasma gondii* (A0A125YJU4, A0A125YYZ6, A0A125YWG5), and *Trypanosoma brucei brucei* (Q4GYY6) reveals residues that differ from mammalian tubulin but are partly conserved among species with subpellicular microtubules (see MSA below). However, as noted above (Response 3), the complex allostery of tubulin makes it difficult to directly link individual residues, which are distributed across the 3D structure, to the mechanical properties or stability of the subpellicular microtubule network. As already noted with respect to oryzalin specificity (lines 163-168), and as we now emphasise in the Discussion with respect to α -2 tubulin (Response 29, lines 403-409), sequence analysis alone may not be sufficient to reveal the full picture of how the properties of subpellicular microtubules are defined across these related yet distinct organisms; additional reconstitution and structural work, alongside *in vivo* mutagenesis studies, will be required.

B) β -tubulin

Rebuttal Figure 5. Multiple sequence alignment of selected protozoan parasite tubulins and pig A) α -tubulin (Q2XVP4) and B) β -tubulin (P02554)

Minor comments:

1. Can the authors include a discussion into other organisms with non-13 protofilament microtubules, and whether they have sequence diversity in similar spots to explain protofilament number diversity? This is reviewed well in Chaaban and Brouhard 2018 (<https://doi.org/10.1091/mbc.e16-05-0271>).

Response 28: Thank you for this interesting suggestion. We have now addressed this point and reference to the Discussion (line 383).

2. Can the authors speculate in the discussion what (if any) structural differences alpha-2 tubulin might provide?

Response 29: The main differences between the sequences of *P. falciparum* alpha-1 and alpha-2 tubulin are in the C-terminal tails and H1-S2 loop regions (See Fig. S2 from Hirst et al, 2022 shown below). As evident from this work, sequence information alone is not sufficient to identify the structural variations between the isoforms. Further these regions are disordered in most microtubule structures determined by cryo-EM. Hence, further structural and functional analyses will be necessary to delineate the consequences of amino acid differences in alpha-2 tubulin, and this is ongoing work in our groups. We have included this point in the Discussion (lines 403-409).

```

α1|Q6ZLZ9|      MREVISIHVGQAGIQVGNACWELFCLEHGIQPDGQMPDQVAVAGGDDAFNTFFSETGAGK 060
α2|Q8IFP3|      MREVISIHVGQAGIQVGNACWELFCLEHGIQPDGQMPDQVAVAGGDDAFNTFFSETGAGK 060
β|Q7KQL5|      MREIVHIQAGQCGNQIGAKFWEVISDEHGDIDPSGTYCGDSDL--QLERVDVVFYNEATGGR 058
                ***:: *:.**.* **:* **::.. *****:*.* .*. : ..*:*:* .*:

α1|Q6ZLZ9|      HVPRCVFVDLEPTVVDEVRTGTYRQLFHPEQLISGKEDAANNFARGHYTIGKEIVDVCLD 120
α2|Q8IFP3|      HVPRCVFVDLEPTVVDEVRTGTYRQLFHPEQLISGKEDAANNFARGHYTIGKEIVDVCLD 120
β|Q7KQL5|      YVPRAILMDLEPGTMDSVRAGPFGQLFRPDNFVFGQTGAGNNWAKGHYTEGAELIDAVLD 118
                :***:::***** .:*.**:* : ***:*::: * : .*.**:*:***** * *::*. **

α1|Q6ZLZ9|      RIRKLDNCTGLQGFLMFSAVGGGTGSGFGCLMLERLSVDYGKSKLNFCCWSPQVSTA 180
α2|Q8IFP3|      RVRKLDNCTGLQGFLMFAVGGGTGSGGLGCLLRLAIDYGKSKLNFCCSWSPQVSTA 180
β|Q7KQL5|      VVRKEAEGCDCLQGFIHTSLGGGTGSGMGTLTLLISKIREEYPRIMETFSVFPSPKVSDT 178
                :** *:.* ***** : :*****:* *::: : * .: .*. :***:* :

α1|Q6ZLZ9|      VVEPYNSVLSHSLLEHTDVAIMLDNEAIYDICRRNLDIERPTYTNLNRLIAQVISSLTA 240
α2|Q8IFP3|      VVEPYNSVLSHSLLEHTDVAIMLDNEAIYDICKKNLDIERPTYTNLNRLIAQVISSLTA 240
β|Q7KQL5|      VVEPYNATLSVHQLVENADEVQVIDNEALYDICFRTLKLTTPTYGDLNHLVSAAMSGVTC 238
                *****:***.*.*:*:* : :*****:***** :*.: *** :**:*:*: :*:*.*

α1|Q6ZLZ9|      SLRFDGALNVDVTEFQTNLVPYPRIHFMLSSYAPVVSAAEKAYHEQLSVSEITNSAFEPAN 300
α2|Q8IFP3|      SLRFDGALNVDVTEFQTNLVPYPRIHFMLSSYAPIISA EKAYHEQLSVSEITNSAFEPAS 300
β|Q7KQL5|      SLRFPGQLNSDLRKLAVNLI PFPRLHFMIGFAPLTSRGSQQYRALTVPELTQQMFDAKN 298
                ***** * ** * : : .**:*:*:*:*: :*:* * . . . * * *:*:* . * :

α1|Q6ZLZ9|      MMAKCDPRHGKYMACLMYRGDVPKDVNAAVATIKTKRTIQFVDWCPTGFKCGINYP 360
α2|Q8IFP3|      MMAKCDPRHGKYMACLMYRGDVPKDVNAAVATIKTKRSIQFVDWCPTGFKCGINYP 360
β|Q7KQL5|      MMCASDPRHGRYLTAAMFRGRMSTKEVDEQMLNVQNKSSYFVEWIPHNTKSSVCDIPP 358
                ** . *****:*:*.* **:* : *:* : : :*:*:* : **:* * . * : : **

α1|Q6ZLZ9|      TVVPGDLAKVMRAVCMISNSTAIAEVFSRMDQKFDLKYAKRAVHWHYVGEEMEEGEFSE 420
α2|Q8IFP3|      TVVPGDLAKVMRAVCMISNSTAIAEVFSRMDQKFDLKYAKRAVHWHYVGEEMEEGEFSE 420
β|Q7KQL5|      KG-----LKMVTFVGNSTAIQEMFKRVSQFTAMFRRKAFHWHYTGEEMDEMEFTE 410
                . : ** : :***** *:*.*:*:* * : :**:*:* .***** * **:*

α1|Q6ZLZ9|      AREDLAALEKDYEEVGIENEAEGEDEGEYEA-- 453
α2|Q8IFP3|      AREDLAALEKDYEEVGIENSDGEGEDEGEY---- 450
β|Q7KQL5|      AESNMNDLVSEYQQYQDATAEEEGEFEEEEEGDVEA 445
                *..:: * .:*:* : : *** * *

```

Rebuttal Figure 6. Sequence alignment of *P. falciparum* alpha-1-, alpha-2-tubulin and beta-tubulin with segments unique to the alpha-1-isoform highlighted in red.

3. I didn't quite understand the significance of TIRF microscopy with regard to the length of microtubules with GMPCPP, particularly given the accompanying EM data:
a. Can the authors include how long the microtubules were polymerised for the TIRF experiments in the methods?

Response 30: To observe the impact of GMPCPP and paclitaxel on *P. falciparum* tubulin, we assembled a reaction *in vitro* and then observed microtubule formation at 37°C using TIRF microscopy. The images depicted in Fig. 4a were taken after 20 min of polymerization; we now provide this information in the figure legend. While EM allows imaging of individual polymers, TIRF microscopy, which is routinely used for imaging microtubules, provides additional information about the size of clusters formed, a property that is reproduced by IRM imaging (Response 14). This has been explained in the Methods section (see pages 26-27). We have also included the details in Fig. 4a legend.

b. Do the authors think “clusters of tubulin” observed are short microtubules, or aggregates? Given the fluorescently labelled tubulin is bovine (understandably), is it possible that the clusters are merely aggregated labelled bovine tubulin, or that the clusters are the result of poor polymerisation between bovine tubulin and plasmodium tubulin? Can the authors clarify these points?

Response 31: We acknowledge this concern and as described in response to Reviewer #1, Response 14, we have repeated these experiments without labelled tubulin by IRM.

Reviewer #5

The authors have managed to generate novel in-vitro reconstituted cryo-EM structures of P.falciparum microtubules under different conditions and systematically compare them to mammalian microtubule standards. By doing this, they have developed a rationale to better understand their differential mechanics and drug susceptibility, based on their structural findings and molecular dynamic simulations. This work represents an advancement in the fields of malaria treatment, P.falciparum biology and microtubule molecular biology and biophysics. The work supports the conclusions drawn, although this reviewer may always take in vitro and in silico studies with a pinch of salt when attempting to draw conclusions about the in cellulo systems (which the authors do not do in the manuscript, as far as I could tell).

Response 32: We note that the data presented in Fig. 5a,b and in Response 25 is *in cellulo* data and shows that our *in vitro* observation of unconventional 15-3 architecture of *P. falciparum* microtubules is found in *P. falciparum* gametocytes. They allow us to link *in vitro* and *in silico* findings with how microtubules behave in the parasites. We have adjusted the relevant section heading (lines 333-334) to make this clearer.

I do not see any reason to prohibit publication of this work, although some minor points could be addressed, mostly for clarity. The methodology is sound, to the extent of my knowledge and regarding the aspects that I am supposed to review. Regarding the cryo-ET standards, a small amount of extra work may have to be carried out (please see comments below).

Fig 1a caption, please specify again the organism of the tubulin used. Also, in Fig 1a, there seems to be a something not too clear as the caption reads “a, ...GTP-paclitaxel (left)” while the label on the image on the left in a says “GDP-paclitaxel”. In Fig 1c the same GDP to GTP change shows up. It can be understood that microtubules will hydrolyze the GTP into GDP, and what is written is correct. Please consider clarifying

this point to make the labels on images and the description on the captions more similar.

Response 33: Apologies for the confusion. The organism information was already included in the Fig. 1a legend. We did not find the text corresponding to GTP-paclitaxel (left) in the legend for Fig. 1a but we have added a clarification in lines 1126-1127 for Fig. 1d legend.

In Extended Data Figure 1 a and EDFig6a, it also would be very informative to have the particle number for each class (or percentage) in the 3D classifications.

Response 34: The percentage of particles per class has now been included in Extended Data Fig. 1a and 6a.

In Fig5 and EDFig5, please add the number of subtomograms used for subtomogram averaging (this cannot be deduced from microtubule number and sampling distance as we do not know the length of each individual microtubule). It would be clear how many particles were picked and used for both in vitro and in vivo averages.

Response 35: For subvolume averaging of the *in vivo* dataset, the number of particles, microtubules and tomograms have now been included in the Methods section (lines 832-833). In this case, all the particles were used to generate a single subvolume average. The processing of the *in vitro* microtubule dataset as depicted in Extended Data Fig. 5 is different because particles from multiple microtubules were averaged independently and hence a range of the particle numbers have been now provided in the figure.

In the Methods section, when the EM movies are recorded, it would be nice to specify their format. I am assuming it is .mrc

Response 36: The EM movies were recorded in the .tiff format. This has been included in the Methods section. Please see lines 610-611

Each structure, independently of its resolution and software used should be deposited (I only see two depositions in the review data, while there are other novel structures coming from this work). There is no report on resolution, angular distribution and other metrics for the in vivo average present in Fig5 and EDFig7. This information seems to be missing also for the in vitro STA of microtubules present in EDFig5. Any EM map, independently of the resolution, should be processed as to be deposited in the EMDB, according to the latest standards (I believe that latest PEET versions do support half-map creation, but they also can be obtained by properly splitting model and motive list files into two random datasets and applying the same .epe particle alignment run to both). RELION can also be used for STA, giving normally better results than PEET and also all sorts of metrics to assess the alignment quality

Response 37: As the reviewer describes, there are two new SPA EM structures determined as part of this work. These have been deposited and metrics included in Table 1. The *in vitro* microtubules were subjected to STA purely to identify their polarity and were not subjected to refinement nor used for structure determination so reconstruction metrics are not relevant for these data - we have updated EDFig5

accordingly to make our data treatment strategy clearer. Finally, the *in vivo* 15-protofilament average in Fig. 5 and ED Fig 7 was redetermined using previously deposited tomograms, informed by our *in vitro*-derived structures. We have now deposited this new *in vivo* average to the EMDB (lines 850-852) and added more metrics to the relevant Methods text (lines 832-833). For reference, we include an FSC curve for the updated volume (below), although given tomography data is already 3D, information concerning angular distribution is less relevant than for SPA, which we have included in ED Figs. 1 and 6. In addition, and as now incorporated, PEET produces the metrics mentioned by the reviewer, and since our existing processing pipeline is established within PEET we are satisfied with its treatment of the data.

Rebuttal Figure 7. FSC plot for the subvolume average of 15-protofilament microtubule from cryo-electron tomography of *P. falciparum* gametocytes. Using a conservative cut-off of 0.5 gives an estimated resolution of ~2.5 nm.

Please indicate the number of micrographs acquired for each dataset in Table 1 (it gives an idea of the microtubule density, particle quality, and other metrics)

Response 38: The number of micrographs have been added to Table 1

In Fig 2, the following can be read: "Further, analysis of the residue composition of the H3 helix 206 revealed an increase in the ratio of non-polar to polar residues at the β - β interface in *P. 207 falciparum* microtubules (Figs. 2c and d; Extended Data Figs. 2a and b)." According to the number of polar and non-polar residues in the figure, the non-polar to polar ratio should decrease, not increase. It would increase if it would be the polar to non-polar ratio. Please only refer to Fig 2 d for β - β interaction.

Response 39: We guess that the organisation of the table and using the ratio of non-polar and polar residues in each tubulin caused confusion here. We now instead

compare numbers of residue types (line 209) and have removed the reference to Fig. 2c.

In Fig 3 g, please include a cross-sectional view of the microtubules in the tomogram to illustrate how easy was it to trace individual protofilaments. ISONET-based filtering may help mitigating missing-wedge artifacts for visualization.

Response 40: We have added cross-sectional views to Extended Data Fig. 5b. While the contribution of software such as ISONET to cryo-ET data treatment has been valuable in a number of samples, typically for lower resolution and more predictable features, it is exactly the unpredictable, variable properties of microtubule ends that are most valuable in shedding light on the relationship between structure and dynamics. As has been the case with other work in the field (Kalutskii M. et al., 2025), we have therefore focused our analysis only on the protofilaments that are least perturbed by the missing wedge.

I would like to suggest the authors, with this or further work, to pursue the experimental elucidation by cryo-EM of the binding site of oryzalin. Also, I believe it is possible to easily reach subnanometric resolution of in vivo microtubules when having access to enough high-quality data from cryo-FIB SEM samples. Although the MD based rigidity assessment is credible, there could be another ways to measure the rigidity of these microtubules: Bending rigidity estimation in partially clamped microtubule gliding assays, optical tweezer approaches or subsection of microtubules to fluid flows in controlled microfluidic chambers.

Response 41: We agree that our current multi-disciplinary study opens many more exciting directions for future investigation - for example, because oryzalin binds to unpolymerized tubulin (now stated explicitly in line 162), ongoing technical developments in cryo-EM would enable direct visualization of oryzalin in complex with unpolymerized parasite tubulin dimers (Wagstaff J.M. et al., 2023). However, this would require extensive sample optimisation given this tubulin's known dynamic properties, and is beyond the scope of the current work. Similarly for the reviewers' other excellent points, we have now briefly highlighted these future opportunities in the Discussion: e.g. lines 441-456 and lines 463-466. See also Response 3 for a discussion of rigidity estimation.

Reviewer #1

The authors strengthened the manuscript by adding the model and measurements of the longitudinal contact surface area, thereby rounding out their argument for the critical change in lateral contacts. A couple of points follow:

*Author Response 3 and 4: While direct testing of stiffness would have significantly strengthened your arguments, I understand it might take months to optimize a gliding assay. Thank you for toning down the language in the manuscript. However, the abstract should also explicitly state that the structure and MD simulations suggest increased stiffness and how *P. fal* microtubules differ in drug susceptibility. The current language implies that the structure proves these features (lines 37-38).*

V2_Response 1. Thank you for this fair comment. We have adjusted the abstract accordingly.

Author Response 6: "However, we have previously shown that at the low concentration used, the presence and source of labelled tubulin does not affect microtubule dynamics (Hirst et al., 2020)." Please mention this statement and reference in the figure legend or methods.

V2_Response 2. Thank you. We have now added this information in the Methods section.

Reviewer #4

The authors have addressed all my points well. As such I would now fully support this for publication. This is an exciting article that will have significance in both the cytoskeleton and malaria communities.

*To address the one remaining point from the authors:
a. Does the GDP-paclitaxel 13-pf reconstruction fit into the 13-pf microtubules observed in the gametocyte?*

Response 25: Yes, it does fit but the resolution of the subvolume average is low and, because there are intrinsically fewer structural features in 13-protofilament microtubules (the protofilaments are straight so there is no moiré pattern), the comparison is less informative. For this reason, we would not propose to include the 13-protofilament microtubule data in the revised manuscript but would be happy to be guided by the reviewer/editor on this point.

The rebuttal figure looks convincing at the resolution, though I agree that this need not to be included.

V2_Response 3. Thanks for this comment. As noted previously, we would be happy to be guided by the editor on this point if they would favour inclusion of this figure.

Reviewer #5

General Comments: It may be beneficial for the manuscript to come up with abbreviations for combinations of words such as 'microtubules', 'brain', 'porcine', 'bovine' and so on.

V2_Response 4. We see the logic of this suggestion, and yet we feel that the inclusion of more abbreviations can cause reading difficulty. Our manuscript already contains quite a few structural and methodological abbreviations, and

since the other reviewers appear satisfied with how the information is presented, we propose not to incorporate additional abbreviations.

The authors have answered all my comments, however there are some points that I find crucial to be revisited. If responses 34 to 35 are not revisited. I would not be able to endorse the publication of this work.

V2_Response 5. Please see detailed responses 8 - 10 below.

I strongly encourage the authors to pursue experimental validation of their MD predictions by other techniques and to consider other technologies such as crosslinking mass spectrometry to investigate the binding site of oryzalin to tubulin

V2_Response 6. We have undertaken a tandem analysis of our experimental cryo-ET data alongside the results of our MD analyses to shed light on the biophysical properties of the parasite microtubules. As noted previously (in V1 Response 41, we agree that our study opens up many more exciting directions for future investigation but new approaches, including crosslinking mass spectrometry both *in vitro* and *in vivo*, are beyond the scope of the current study.

Fig 1a caption, please specify again the organism of the tubulin used. Also, in Fig 1a, there seems to be a something not too clear as the caption reads “a, ...GTP-paclitaxel (left)” while the label on the image on the left in a says “GDP-paclitaxel”. In Fig 1c the same GDP to GTP change shows up. It can be understood that microtubules will hydrolyze the GTP into GDP, and what is written is correct. Please consider clarifying this point to make the labels on images and the description on the captions more similar.

Response 33: Apologies for the confusion. The organism information was already included in the Fig. 1a legend. We did not find the text corresponding to GTP-paclitaxel (left) in the legend for Fig. 1a but we have added a clarification in lines 1126-1127 for Fig. 1d legend.

33: Yes, it should always be clear in the text to which organism the structures correspond together with the experimental conditions used (in silico, in vitro, in cellulo, etc). When I cite the following piece of your manuscript: “a,[...]GTP-paclitaxel (left)”, this corresponds to Figure 4 caption, not Figure 1, mea culpa. Still, while the label on the image on the left in a says “GDP-paclitaxel”. The same apparent mistake can be found in Figure 4 c. I am still not sure if this change in nomenclature is due to the GTP hydrolysis process or just a mislabeling. Please check

V2_Response 7. i) Thanks for clarifying that the query concerned Figure 4a, which indeed wasn't clear and is now corrected. ii) The same issue caused confusion in the legend for Figure 4c and we have adjusted the text to be more consistent.

In Extended Data Figure 1 a and EDFig6a, it also would be very informative to have the particle number for each class (or percentage) in the 3D classifications.

Response 34: The percentage of particles per class has now been included in Extended Data Fig. 1a and 6a.

34: Thanks for adding the percentages. I still have a serious concern about the 0% classes in Extended Data Fig 6 a. It is very confusing, if there are 0% of the particles in a class, how where the class averages generated for that class?

V2_Response 8. Apologies for the confusion – there are very few particles in the

11-3, 12-3 and 14-4 classes in this dataset and previously including only 1 decimal place caused them to be set to 0%. We have now presented these values to 2 decimal places, which we hope will clarify this point.

When comparing them to Extended Data Figure 1a the images do not match.

V2_Response 9. It is not totally clear what is meant by “do not match” between these figure panels, but as explained in the manuscript, microtubules polymerized under different conditions exhibit different distributions of architectures so these are not identical between datasets. In addition, the microtubules depicted in Extended Data Figure 1a have a kinesin motor domain fiducial marker bound to enable particle alignment, while those in Extended Data Figure 6a do not. We have now specified this in Extended Data (now Supplementary) Figure 1 legend.

In Fig5 and EDFig5, please add the number of subtomograms used for subtomogram averaging (this cannot be deduced from microtubule number and sampling distance as we do not know the length of each individual microtubule). It would be clear how many particles were picked and used for both in vitro and in vivo averages.

Response 35: For subvolume averaging of the *in vivo* dataset, the number of particles, microtubules and tomograms have now been included in the Methods section (lines 832-833). In this case, all the particles were used to generate a single subvolume average. The processing of the *in vitro* microtubule dataset as depicted in Extended Data Fig. 5 is different because particles from multiple microtubules were averaged independently and hence a range of the particle numbers have been now provided in the figure.

35: The changes in lines 832-833 are insufficient to allow the reader to easily understand the total microtubule length used for STA. The initial sampling distance along the microtubules remains undisclosed as it is veiled in the following text: “...initial coordinates were generated by interpolating between these points using scripts based on TEMPy102”. At least it should be mentioned how many tubulin-repeats (~4nm) where picked. Furthermore, after duplicate removal and similar processing, the final number of particles is not disclosed in the corresponding Materials and Methods section. Please include this information.

V2_Response 10. Apologies – we have further updated the Methods text to include more detailed information.

In Extended Data Figure 5b, the “brain” label could read ‘bovine’ or something like that to ease the interspecies comparison for the reader.

V2_Response 11. As endorsed by Reviewer 1, we have simplified our nomenclature throughout the manuscript and feel it is important to maintain this consistency throughout, including in the figure panel.

Extended Data Figure 5b has now particle number estimations. However, the image under the label “Average 3D volume” comes from a specific average, with specific particle numbers, please disclose that number.

V2_Response 12. We have updated Extended Data (now Supplementary) Fig. 5a (which is the panel the reviewer is referring to) to include this information.

In Fig 3 g, please include a crosssectional view of the microtubules in the tomogram to

illustrate how easy was it to trace individual protofilaments. ISONET-based filtering may help mitigating missing-wedge artifacts for visualization.

Response 40: We have added cross-sectional views to Extended Data Fig. 5b. While the contribution of software such as ISONET to cryo-ET data treatment has been valuable in a number of samples, typically for lower resolution and more predictable features, it is exactly the unpredictable, variable properties of microtubule ends that are most valuable in shedding light on the relationship between structure and dynamics. As has been the case with other work in the field (Kalutskii M. et al., 2025), we have therefore focused our analysis only on the protofilaments that are least perturbed by the missing wedge.

40: I do appreciate that the authors included the microtubule crosssections in Extended Data Figure 5b. However, the crosssections do not correspond to the protofilaments in the “curved ends”.

V2_Response 13. In fact, what is depicted is a projection from the microtubule end to provide an overview of its shape, including the splayed protofilaments, depicted as green lines. However, the legend wasn't completely clear on that point. We have now added views of the tomogram sections with and without filament traces, included a key and corrected the legend in Supplementary Fig. 5b.

Could the authors show a crosssection of these regions to illustrate how easy it was to trace the protofilaments? Was a combination of crosssections and longitudinal sections used to trace the protofilaments?

V2_Response 14. As described in the Methods, we followed the approach first rigorously described and tested by McIntosh et al (PMID: 29794031), which involves only using longitudinal sections to trace protofilaments; we have added further clarification in the Methods text on this point. Since this is extensively described in the original publication and is well established in the microtubule field, we considered it a redundant addition in our manuscript. We now include an example from our own data below and would be pleased to follow editorial guidance about its inclusion in the manuscript.

Rebuttal_V2 Figure 1. Exemplar tomogram slices of *P. falciparum* paclitaxel-stabilized microtubule ends used for protofilament tracing, rotated around the microtubule longitudinal axis (top) with respective protofilament traces (bottom) Scale bars=20nm.

I would like to suggest the authors, with this or further work, to pursue the experimental elucidation by cryo-EM of the binding site of oryzalin. Also, I believe it is possible to easily reach subnanometric resolution of in vivo microtubules when having access to enough high-quality data from cryo-FIB SEM samples. Although the MD based rigidity assessment is credible, there could be another ways to measure the rigidity of these microtubules: Bending rigidity estimation in partially clamped microtubule gliding assays, optical tweezer approaches or subjection of microtubules to fluid flows in controlled microfluidic chambers.

Response 41: We agree that our current multi-disciplinary study opens many more exciting directions for future investigation - for example, because oryzalin binds to unpolymerized tubulin (now stated explicitly in line 162), ongoing technical developments in cryo-EM would enable direct visualization of oryzalin in complex with unpolymerized parasite tubulin dimers (Wagstaff J.M. et al.,2023). However, this would require extensive sample optimisation given this tubulin's known dynamic properties, and is beyond the scope of the current work. Similarly for the reviewers' other excellent points, we have now briefly highlighted these future opportunities in the Discussion: e.g. lines 441-456 and lines 463-466. See also Response 3 for a discussion of rigidity estimation.

41: I would like to add that cross-linking mass spectrometry can shed light onto the tubulin binding site of oryzalin also in vivo

V2_Response 15. Please see response 5.